# Plasma metabolic profile reveals signatures of maternal health during gestational hypertension and preeclampsia without and with severe features

Julyane N. S. Kaihara[1], Fabio Rogerio de Moraes[2], Priscila Rezeck Nunes[1], Marco G. Alves[3], Ricardo C. Cavalli[4], Ljubica Tasic[5], Valeria Cristina Sandrim[1]*

1 Department of Biophysics and Pharmacology, Institute of Biosciences of Botucatu, Sao Paulo State University (UNESP), Botucatu, SP, Brazil, 2 Multiuser Center for Biomolecular Innovation, Department of Physics, Institute of Biosciences, Languages and Exact Sciences, Sao Paulo State University (UNESP), Sao Jose do Rio Preto, SP, Brazil, 3 Institute of Biomedicine and Department of Medical Science (iBiMED), University of Aveiro, Aveiro, Portugal, 4 Department of Gynecology and Obstetrics, Faculty of Medicine of Ribeirao Preto, University of Sao Paulo (USP), Ribeirao Preto, SP, Brazil, 5 Department of Organic Chemistry, Institute of Chemistry, State University of Campinas (UNICAMP), Campinas, SP, Brazil

* valeria.sandrim@unesp.br

**Data Availability Statement:** All relevant data are within the manuscript and its Supporting Information files.

## Abstract

Preeclampsia, a pregnancy-specific syndrome, poses substantial risks to maternal and neonatal health, particularly in cases with severe features. Our study focuses on evaluating the impact of low molecular weight metabolites on the intricate mechanisms and pathways involved in the pathophysiology of preeclampsia when severe features are present. We aim to pinpoint the distinct metabolomic profile in maternal plasma during pregnancies affected by hypertensive disorders and to correlate the metabolite levels with the clinical characteristics of the study cohort. A total of 173 plasma samples were collected, comprising 36 healthy pregnant women (HP), 52 patients with gestational hypertension (GH), 43 with preeclampsia without (PE–), and 42 with severe features (PE+). Nuclear magnetic resonance spectroscopy and metabolite identification were conducted to establish the metabolomic profiles. Univariate and chemometric analyses were conducted using MetaboAnalyst, and correlations were performed using GraphPad Prism. Our study unveils distinct metabolomic profiles differentiating HP women, patients featuring GH, and patients with PE–and PE+. Our analysis highlights an increase in acetate, *N,N*-dimethylglycine, glutamine, alanine, valine, and creatine levels in the PE+ group compared to the HP and GH groups. The PE+ group exhibited higher concentrations of *N,N*-dimethylglycine, glutamine, alanine, and valine compared to the PE–group. Moreover, elevated levels of specific metabolites, including *N,N*-dimethylglycine, alanine, and valine, were associated with increased blood pressure, worse obstetric outcomes, and poorer end-organ function, particularly renal and hepatic damage. Metabolomic analysis of PE+ individuals indicates heightened disturbances in nitrogen metabolism, methionine, and urea cycles. Additionally, the exacerbated metabolic disturbance may have disclosed renal impairment and hepatic dysfunction, evidenced by elevated levels of creatine and alanine. These findings not only contribute novel insights but

**Funding:** This work was supported by the Coordenação de Aperfeiçoamento de Pessoal de Nível Superior (CAPES) [grant number 88887.806462/2023-00 (JNSK)], the Conselho Nacional de Desenvolvimento Científico e Tecnológico (CNPq) [grant numbers 308079/2021-3 (LT) and 308504/2021-6 (VCS)], and the Fundação de Amparo à Pesquisa do Estado de São Paulo (FAPESP) [grant numbers 2018/24069-3 (LT), 2019/07230-8, 2021/12010-7 (VCS), and 2023/08897-1 (JNSK)]. iBiMED is funded by Fundação para a Ciência e a Tecnologia (FCT) [UIDP/04501/2020 and UIDB/04501/2020 (MGA)]. The founders had no role in study design, data collection and analysis, decision to publish, or preparation of the manuscript. There was no additional external funding received for this study".

**Competing interests:** The authors have declared that no competing interests exist.

also provide a more comprehensive understanding of the pathophysiological mechanisms at play in cases of PE+.

## Introduction

Preeclampsia (PE) is a pregnancy-specific syndrome with profound implications for both maternal and neonatal health. Its association with heightened risks of morbidity and mortality often necessitates immediate intervention to prevent grave short- and long-term complications [1]. Globally, PE is estimated to affect 2 to 8% of all pregnancies [2], contributing to the death of more than 500,000 fetuses and newborns, along with the loss of over 70,000 mothers annually [1, 3]. In Brazil, a staggering 37% of maternal deaths stem from obstetric complications linked to PE [4].

PE features hypertension and proteinuria, emerging during pregnancy or the postpartum period. In the absence of proteinuria, PE is diagnosed based on hypertension accompanied by symptoms indicative of end-organ damage [2]. Furthermore, PE is stratified into categories without or with severe features, determined by the severity of symptoms [2, 3]. Instances of PE with severe features (PE+) are closely linked to unfavorable maternal and fetal outcomes, often necessitating iatrogenic premature delivery when the health of either is jeopardized [5]. As a result, premature delivery amplifies the probability of adverse neonatal outcomes, encompassing respiratory and gastrointestinal complications, neurological and developmental impairments, and an elevated risk of neonatal mortality [6–8]. Characterized by a heterogeneous pathophysiology, PE emerges as a complex and multifactorial disorder. Patients with PE exhibit a generalized endothelial dysfunction, a state marked by the compromised integrity of endothelial cells, stemming from inadequate release of placental factors [9, 10]. This impaired endothelial function stands as the driving force behind the clinical manifestations of PE, influencing the levels of both vasodilator and vasoconstrictor factors. Notably, this involves a reduction in nitric oxide (NO) and prostacyclin levels and an elevation in endothelin levels [9]. The intricate interplay of these factors underscores the complexity of PE, shedding light on its intricate web of causative elements and clinical consequences.

Metabolomics delves into the comprehensive exploration of endogenous metabolites within biological specimens, encompassing biofluids, cells, and tissues. Metabolites, vital components of an organism's metabolism, play dual roles as both substrates and products in biochemical reactions. Crucial for cellular functions such as energy generation and signal transduction, a profound understanding of these compounds becomes imperative for unraveling the molecular pathophysiology of diverse diseases [11]. Previous studies performed using serum metabolomics analyses have identified numerous altered metabolites in patients with severe features of PE compared to control groups. Notable among these are arginine, alanine, aspartate, glutamate, and histidine, along with other amino acid metabolites [12]. Additionally, oxidized phospholipids [13] and 2-hydroxybutyric acid [14] have also been identified. These metabolites have been linked to vasoconstriction and endothelial dysfunction [9], the development of metabolic risk [12], and potential associations with oxidative stress [12–14]. Moreover, additional research has detected alterations in metabolites and pathways between placentas from pregnancies experiencing PE+ and normotensive pregnancies [15–18]. The technique known as $^1$H nuclear magnetic resonance ($^1$H-NMR) spectroscopy is a widely used approach for accurately quantifying metabolites [19]. This methodology offers advantages including high reproducibility and does not require harsh sample processing either before or during the

experiments. This is important for analyzing specific metabolites and sample preservation [20]. Surprisingly, none of these studies have considered pregnancies experiencing PE without severe features (PE–), PE+, or gestational hypertension (GH), utilizing [1]H-NMR spectroscopy to compare the metabolomic profiles of plasma among patients with these hypertensive disorders. Studying the levels of metabolites across the different hypertensive disorders of pregnancy may provide deeper insights into the underlying mechanisms of the interrelated conditions of GH, PE–, and PE+.

In this study, our primary objective was to unravel the profile of low molecular weight metabolites, shedding light on the intricate mechanisms and pathways underpinning the pathophysiology of PE+. Our investigation aimed to discern the altered metabolomic landscape within maternal plasma in pregnancies affected by hypertensive disorders, which was successfully achieved by conducting [1]H-NMR experiments and subsequent bioinformatics analysis on a cohort encompassing healthy pregnant (HP) women, individuals with GH, and patients with PE–or PE+. Subsequently, we investigated the presence of correlations between the metabolite concentrations and the clinical characteristics of the study cohort.

## Materials and methods

### Study population

A study flow chart is presented in Fig 1. This cross-sectional case-control study was approved by the Institutional Review Board from the Ribeirao Preto Medical School of the University of Sao Paulo, Brazil (FMRP/USP, protocol code CAAE: 37738620.0.0000.5440, on October 19, 2020). The procedures were conducted in compliance with both the Declaration of Helsinki and the applicable institutional policies and regulations for studies involving human subjects. At clinical attendance, participants were recruited voluntarily, and all provided written informed consent. If there were pregnant women under the age of 18, their parents or legal guardians were required to give written consent. For research purposes, we accessed the clinical data of the archived samples on July 13, 2023. No information that could identify individual participants was accessed during or after data collection. All data were fully encoded and anonymized to ensure the privacy and confidentiality of the subjects' information. This study enrolled a total of 173 participants: HP ($n$ = 36) women from the Reference Center for Women's Health of Ribeirao Preto (MATER), and GH ($n$ = 52), PE–($n$ = 43), and PE+ ($n$ = 42) patients from the Department of Obstetrics and Gynecology, University Hospital of the Ribeirao Preto Medical School (HCFMRP).

The participants were included in the PE groups based on the criteria outlined by the American College of Obstetrics and Gynecology (ACOG) [2]. The diagnostic criteria of PE included hypertension (systolic blood pressure $\geq$ 140 mmHg and/or diastolic blood pressure $\geq$ 90 mmHg) in women previously normotensive after 20 weeks of gestation accompanied by proteinuria. In the absence of proteinuria, hypertension with symptoms of damage to the end-organs such as platelets, kidneys, liver, lungs, brain, and eyes were considered. The severity of PE was determined by pregnancies with one or more of the following features, as defined by the guidelines of ACOG and the ISSHP (International Society for the Study of Hypertension in Pregnancy) [2, 21]: systolic blood pressure $\geq$ 160 mmHg and/or diastolic blood pressure $\geq$ 110 mmHg, thrombocytopenia, progressive renal insufficiency, hepatic impairment, pulmonary edema, and symptoms affecting the brain or vision. A new onset of hypertension (blood pressure $\geq$ 140/90 mmHg) after 20 weeks of gestation without proteinuria and/or without the aforementioned symptoms of end-organ damage was defined as GH [2]. The HP group served as control and consisted of pregnant women who had no previous medical conditions or antenatal complications. Pregnancies associated with fetal or hemostatic

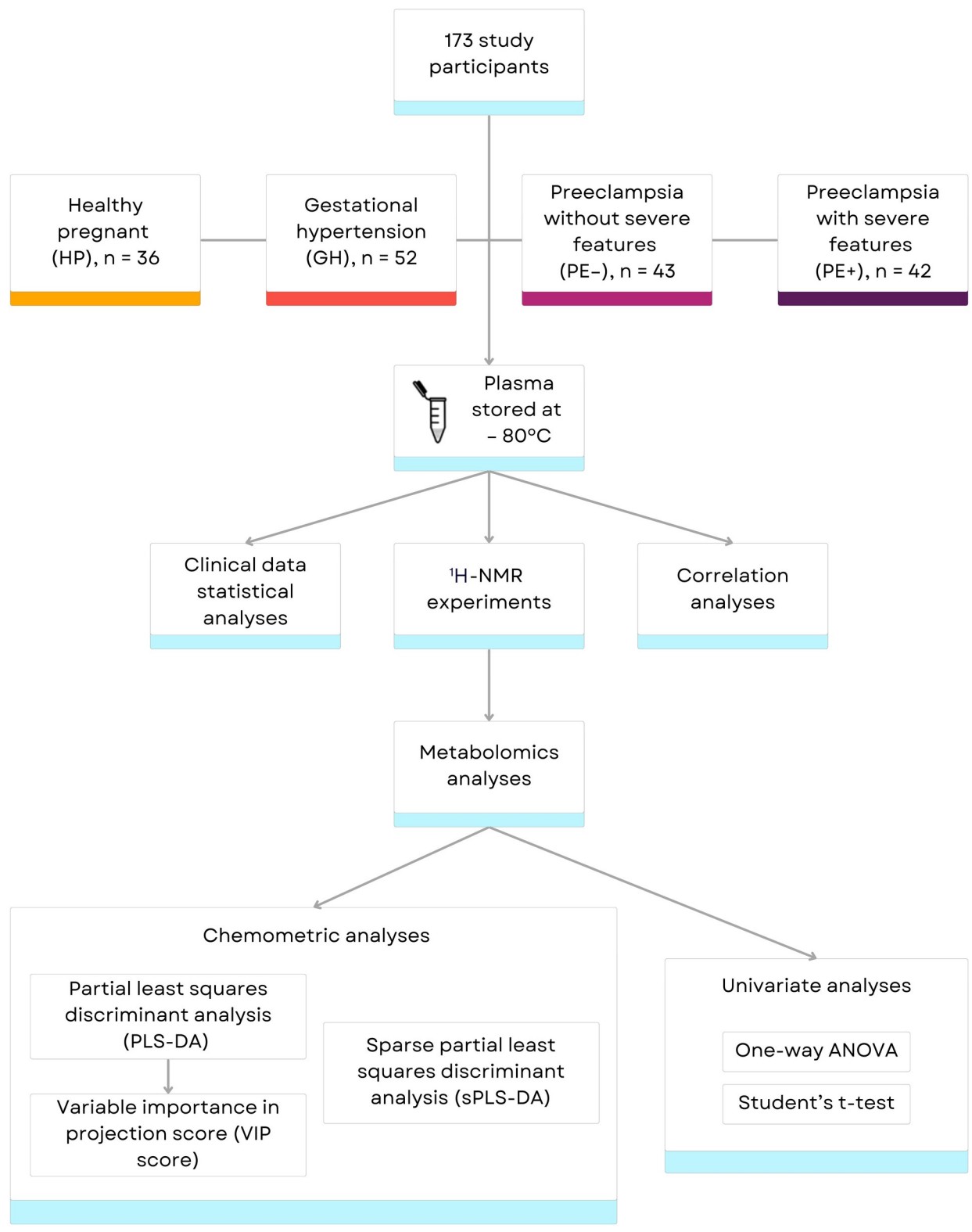

**Fig 1. Study flow chart.**

abnormalities, chronic hypertension, cancer, and diabetes mellitus, as well as those with cardiovascular conditions, and autoimmune, renal, or hepatic diseases, were excluded from this study. Data on the demographic and clinical information of pregnant women and their newborns were collected from medical records, alongside the results of routine laboratory tests. Samples from all groups were collected during 22–42 weeks of pregnancy.

## Plasma sampling

Upon diagnosis, the blood of preeclamptic women was collected, while the blood of pregnant women without complications and GH patients was collected during routine clinical attendance. Maternal venous blood samples were collected into sterile tubes with 10 U/mL EDTA (Becton Dickinson-BD Vacutainer; BD Biosciences, Franklin Lakes, NJ, USA). The samples were centrifuged, and the plasma (supernatant) was separated. The aliquots obtained were stored at $-80°C$ until the NMR experiments were conducted for the present study.

## Metabolomics: Nuclear magnetic resonance spectroscopy (NMR)

All plasma samples were thawed at $4°C$. Next, 200 μL of deuterated water ($D_2O$) was added to each 300 μL of individual plasma samples. Subsequently, the resulting solution was transferred into a 5 mm nuclear magnetic resonance tube. Metabolomics runs were executed on a Bruker Ascend™ 600 MHz spectrometer (Bruker BioSpin, Rheinstetten, Germany) equipped with a room-temperature TXI-Z triple inverse probe for acquiring $^1H$-NMR spectra. The one-dimensional $^1H$-NMR spectra were obtained through NOESY (Nuclear Overhauser Effect Spectroscopy) presaturation experiments to suppress water signals. The parameters included a recycle delay of 4 s, mixing time of 150 ms, 128 scans, and the collection of 64k complex data points within a spectral width of 20 ppm. Post-acquisition, the free induction decay measurement (FIDs) was multiplied by an exponential function with 1 Hz line-broadening, followed by Fourier transformation. CPMG (Carr-Purcell-Meiboom Gill) pulse sequence was also employed with a 200 loop for $T_2$ filter, effectively suppressing macromolecule signals. This sequence shared similar parameters with the NOESY 1D experiment, encompassing a recycle delay of 4 s, 128 scans, and 64k complex data points. For metabolite identification, 2-dimensional $^1H$-$^1H$ TOCSY (Total Correlation Spectroscopy) and $^1H$-$^{13}C$ HSQC (Heteronuclear Single Quantum Coherence) spectra were acquired from randomly selected plasma samples. The data processing involved the use of TopSpin software version 3.2 (Bruker BioSpin, Rheinstetten, Germany), including Fourier transformation on FID signals, as well as phase and baseline corrections. Further refinement, including reference setting (lactate signal, 3H, δ 1.33, $J = 7.0$ Hz, doublet) and manual inspection of the spectra was carried out in MestReNova 14.2.2 (2021 Mestrelab Research S.L.). Metabolite identification spanned amino acids, ketone bodies, fatty acids, and polar molecules facilitated by Chenomx NMR Suite software version 9.02 (Chenomx Inc., Edmonton, AB, Canada). This involved comparing peak spectra, chemical shifts (in ppm), and scalar couplings ($J$) with the Spectral Reference Library of the software, aligning with the conditions under which the data was acquired, including pH, temperature, and NMR field strength. When signals were not cataloged in the Chenomx software database, the Human Metabolome Database (HMDB) [22] was referenced.

## Statistical and metabolomics analyses

The clinical, demographic, and biochemical characteristics of the participants in the study were subjected to normality tests to evaluate their distributions. Statistical tests, including One-way ANOVA, Welch's ANOVA, and Kruskal-Wallis tests, were used to analyze differences between all groups, as appropriate. For pairwise group comparisons, post-hoc tests such

as Bonferroni or Dunn's were conducted for parametric and non-parametric data, respectively. Categorical variables were compared using Chi-square ($\chi^2$). Pearson or Spearman correlation tests were employed to evaluate the relationship between creatinine levels measured in the clinical setting and those determined via $^1$H-NMR. Additionally, correlation analyses were conducted between the clinical characteristics of all the study population and the metabolite concentrations. GraphPad Prism version 9.5 (GraphPad Software Inc., San Diego, CA, USA) was used to perform these analyses. A significance level of $p < 0.05$ was considered for statistical analysis.

Metabolomics data were Pareto scaled before further analysis. Chemometric analyses comprising partial least squares discriminant analysis (PLS-DA) and sparse partial least squares discriminant analysis (sPLS-DA) were employed. A variable importance in projection (VIP) score was generated, visually representing the significance of specific metabolites in differentiating the experimental groups. A threshold higher than 1.0 was considered significant. To compare differences in plasma metabolite concentrations between experimental groups, we conducted univariate analyses using One-way ANOVA for comparisons between the four groups, and Student's unpaired t-tests for comparisons between two groups, both with a false discovery rate (FDR) value of 0.05. All chemometric and univariate analyses were conducted using the online software MetaboAnalyst, versions 5.0 and 6.0 (http://www.metaboanalyst.ca/).

## Results

### Characteristics of the study population

The clinical, demographic, and biochemical characteristics of the 173 study participants are presented in Table 1, and the raw data for this cohort are available in S1 Table. The average maternal age, frequency of primiparity, heart rate, sodium, potassium, and total bilirubin were similar across all experimental groups (all $p > 0.05$). GH and PE–had higher BMI (body mass index) measurements than HP and PE+ ($p < 0.0010$). There was a significant difference in the gestational age at sampling between the PE+ group and the other experimental groups ($p < 0.0010$). The GH, PE–, and PE+ higher levels of systolic and diastolic blood pressure levels were significant when compared to those of HP ($p < 0.0010$), and PE+ also had higher systolic blood pressure than GH ($p < 0.0010$), as expected. Platelet count was significantly lower in the PE+ group compared to the GH group ($p = 0.0045$). PE–and PE+ urea levels were higher than GH, and PE+ urea levels were also higher than PE–($p < 0.0010$). The liver enzyme aspartate transaminase ($p < 0.0010$) and creatinine levels ($p < 0.0010$) were elevated in the PE+ group compared to GH and PE–. PE–and PE+ 24-hour proteinuria–excretion of proteins in the urine over a 24-hour period–and direct bilirubin levels were significantly higher than GH ($p < 0.0010$ and $p = 0.0060$, respectively). The gestational age at delivery was lower in the PE + group compared to all the other groups ($p < 0.0010$). Subjects from the PE+ group had newborns with lower weight ($p < 0.0010$) than all the other groups. The incidence of APGAR scores $< 7$ during the initial assessment was greater in the PE+ group compared to HP ($p = 0.0092$). There was no difference in the incidence of APGAR scores $< 7$ at the secondary assessment comparing all groups ($p = 0.2075$).

### Metabolomics results

Metabolomic analyses of the plasma samples yielded the identification of a total of 19 metabolites: 3-hydroxybutyrate (HMDB0000011), acetate (HMDB0000042), acetone (HMDB0001659), alanine (HMDB0000161), citrate (HMDB0000094), creatine (HMDB0000064), creatinine (HMDB0000562), formate (HMDB0000142), glucose

**Table 1. Clinical, demographic, and biochemical characteristics of healthy pregnant (HP) women, gestational hypertension (GH) subjects, and preeclampsia patients without severe features (PE–) and with severe features (PE+)[a].**

| Parameters | HP | GH | PE– | PE+ | *p*-value |
|---|---|---|---|---|---|
| **Age (years)** | 24.5 (5.7) | 26.1 (6.2) | 26.9 (6.3) | 28.0 (8.0) | 0.1282 |
| **Primiparity (%)** | 55.6 | 44.4 | 37.2 | 40.5 | 0.3964 |
| **BMI (kg/m$^2$)** | 27.5 (4.5) | 35.7 (7.1)[b] | 35.2 (5.8)[b] | 30.2 (6.3)[cd] | **< 0.0010** |
| **GAS (weeks)** | 37.0 [31.0–39.0] | 37.0 [24.0–42.0] | 37.0 [26.0–42.0] | 32.0 [22.0–41.0][bcd] | **< 0.0010** |
| **SBP (mmHg)** | 110.0 [90.0–140.0] | 130.0 [100.0–190.0][b] | 134.0 [110.0–180.0][b] | 140.0 [106.0–200.0][bc] | **< 0.0010** |
| **DBP (mmHg)** | 70.0 [50.0–90.0] | 80.0 [70.0–102.0][b] | 90.0 [60.0–120.0][b] | 90.0 [68.0–130.0][b] | **< 0.0010** |
| **HR (bpm)** | 80.0 [70.0–95.0] | 80.0 [68.0–100.0] | 80.0 [66.0–96.0] | 80.0 [60.0–101.0] | 0.2362 |
| **Platelets (×10$^3$/mm$^3$)** | ND | 235.0 (61.5) | 223.5 (70.1) | 190.3 (64.9)[c] | **0.0045** |
| **Sodium (mmol/L)** | ND | 139.3 (3.4) | 138.1 (2.3) | 137.9 (3.6) | 0.0604 |
| **Potassium (mmol/L)** | ND | 4.1 (0.4) | 4.0 (0.4) | 4.0 (0.4) | 0.8140 |
| **Urea (mg/dL)** | ND | 12.9 (3.7) | 17.2 (5.8)[c] | 24.9 (10.4)[cd] | **< 0.0010** |
| **AST (U/L)** | ND | 15.0 [8.0–28.0] | 16.0 [9.0–32.0] | 20.0 [5.0–193.0][cd] | **< 0.0010** |
| **Creatinine (mg/dL)** | ND | 0.6 [0.4–1.0] | 0.6 [0.4–0.9] | 0.7 [0.5–2.9][cd] | **< 0.0010** |
| **24 h Pr (mg/24h)** | ND | 161.0 [48.0–259.0] | 410.0 [304.0–3889.0][c] | 1163.0 [302.0–6768.0][c] | **< 0.0010** |
| **Total Bilirubin (mg/dL)** | ND | 0.4 [0.1–0.8] | 0.3 [0.1–1.2] | 0.3 [0.1–1.0] | 0.1847 |
| **Direct Bilirubin (mg/dL)** | ND | 0.1 [0.0–0.4] | 0.1 [0.0–0.5][c] | 0.1 [0.0–0.5][c] | **0.0060** |
| **GAD (weeks)** | 39.0 [37.0–42.0] | 39.0 [36.0–42.0] | 38.0 [36.0–42.0] | 34.0 [22.0–41.0][bcd] | **< 0.0010** |
| **Newborn Weight (g)** | 3200.0 (403.0) | 3309.0 (448.5) | 3198.0 (418.8) | 1655.0 (720.1)[bcd] | **< 0.0010** |
| **APGAR Score 1 (1 min) < 7 (%)** | 0.0 | 10.0[b] | 8.0[b] | 11.0[b] | **0.0092** |
| **APGAR Score 2 (5 min) < 7 (%)** | 0.0 | 0.0 | 1.0 | 2.0 | 0.2075 |

24 h Pr, 24 h proteinuria; AST, aspartate transaminase; BMI, body mass index; DBP, diastolic blood pressure; GAD, gestational age at delivery; GAS, gestational age at sampling; HR, heart rate; ND, not determined; SBP, systolic blood pressure.

[a] Values are given as mean (standard deviation) for parametric variables; median [minimum-maximum] for non-parametric variables; and frequency (as percentage) for categorical variables. One-way ANOVA, Welch's ANOVA, Kruskal-Wallis, or Chi-square ($\chi^2$) tests were applied as appropriate, followed by correction for multiple comparisons. Categorical variables were compared by contingency table. Bold values are significant *p*-values.

[b] $p < 0.05$ versus HP

[c] $p < 0.05$ versus GH

[d] $p < 0.05$ versus PE–

(HMDB0000122), glutamine (HMDB0000641), histidine (HMDB0000177), isoleucine (HMDB0000172), lactate (HMDB0000190), leucine (HMDB0000687), *N*,*N*-dimethylglycine (HMDB0000092), tyrosine (HMDB0000158), valine (HMDB0000883), resonance at 2.996 ppm (an unidentified metabolite), and *N*-acetylglycoproteins (corresponding to the superimposition of signals from molecules containing *N*-acetyl groups). S1 and S2 Figs showcase all NMR spectra superposed based on each specific category (0.0–9.0, and insets 6.5–8.5 ppm). Additionally, the presentation includes a correlation map for the ${}^1$H-${}^1$H NMR TOCSY experiment, highlighting the peaks and their corresponding metabolite identifications.

S3 Fig illustrates the positive correlation between creatinine levels quantified in routine clinical testing and those determined via ${}^1$H-NMR in the PE+ group. A significant correlation was observed between the two sets of creatinine levels (S3A Fig, rs = 0.3568, *p* = 0.0204), and this correlation remained significant even after excluding the evident outlier (S3B Fig, r = 0.4150, *p* = 0.0070). The results demonstrate that both creatinine measurement techniques yield consistent results, thereby underscoring the reliability of both methods.

Metabolomic analysis was used to determine disparities in plasma metabolite concentrations among the HP, GH, PE–, and PE+ groups. Here, we focused our analysis on the PE

+ group, driven by the clinical significance and severity of this condition. The supplementary file contains the remaining comparisons. Fig 2 illustrates the findings, with Fig 2A displaying a 2-dimensional plot acquired by sPLS-DA. Each data point represents an analyzed sample, and the colored ellipsoids outline the experimental groups at a 95% confidence interval. The first principal component, located on the X-axis, accounted for 27.8% of the general variation within the groups. The second principal component, located on the Y-axis, explains 12.3% of the variation. Fig 2B presents important metabolites for differentiating the groups, such as alanine, acetate, glutamine, glucose, and N,N-dimethylglycine. The VIP score increases along the X-axis, implying a greater role in group separation. Furthermore, the squares and heatmap on the right side of the VIP plot indicate that the concentration of these metabolites was higher in the PE+ group. Consistent with the chemometric analysis, a univariate analysis has uncovered twelve significant metabolites, namely: acetate, N,N-dimethylglycine, alanine, creatinine, citrate, glutamine, creatine, histidine, 2.996, valine, N-acetylglycoproteins, and leucine (displayed in Fig 2C), all of which had significantly different levels in at least one patient group.

A comparison between HP and PE+ was performed (Fig 3). Applying sPLS-DA on the HP and PE+ data indicates that the first principal component was responsible for 29.5% of the total variation between the groups, and the second component accounted for 13.4% (Fig 3A and 3B). The 2-dimensional and 3-dimensional plots demonstrate a greater dispersion among the studied individuals in the PE+ group. The VIP score plot displays alanine, acetate, glutamine, glucose, N,N-dimethylglycine, and valine as important metabolites for determining group separation (Fig 3C). Similar metabolites were significant in univariate analysis and also included citrate, 2.996, 3-hydroxybutyrate, creatine, and acetone (Fig 3D). In this analysis, glucose, 3-hydroxybutyrate, and acetone were different among the HP and PE+ groups and were not different in the previous global analysis when comparing all the groups.

The metabolomics analysis with GH and PE+ groups (Fig 4) indicates that the first principal component was responsible for 28.9% of the total variation between the groups, whereas the second component represents 16.7% (Fig 4A and 4B). Alanine, N-acetylglycoproteins, acetate, and glutamine emerged as pivotal metabolites in discerning distinct groups (Fig 4C). Consistent with these outcomes, the univariate analysis also indicated alanine, N-acetylglycoproteins, acetate, and glutamine, as well as creatinine, N,N-dimethylglycine, creatine, leucine, acetone, and valine as significantly elevated metabolites in the PE+ group (Fig 4D). In contrast to the comprehensive analysis involving all groups, this specific univariate analysis pinpointed a notable distinction in the levels of acetone between the GH and PE+ groups.

Fig 5 displays the results of the chemometric analysis conducted on PE−and PE+ groups. Component 1 explained 33.8% of the overall variation between the two groups, and component 2 explained 8.7% (Fig 5A and 5B). According to the VIP score plot, the metabolites with the most significant contribution to group separation were N-acetylglycoproteins, alanine, glutamine, and isoleucine (Fig 5C). Fig 5D shows that there were significant differences between the two groups in terms of N-acetylglycoproteins, alanine, and glutamine, as well as N,N-dimethylglycine, histidine, creatine, and valine. In this univariate analysis, acetate, creatinine, histidine, 2.996, and leucine were not different between the PE−and PE+ groups as in the previous analysis with all the groups identified.

We performed both chemometric and univariate analyses to compare HP vs GH, HP vs PE−, and GH vs PE−groups as demonstrated in S4–S6 Figs, respectively. The GH group had significantly higher concentrations of citrate, N,N-dimethylglycine, and 2.996 compared to HP, while creatinine was significantly higher in the HP group. The PE−group exhibited higher concentrations of acetate in contrast to both HP and GH groups. Furthermore, the PE−displayed higher levels of citrate compared to the HP group. Notably, increased levels of histidine and decreased levels of creatinine were observed in the PE−group when compared to the GH

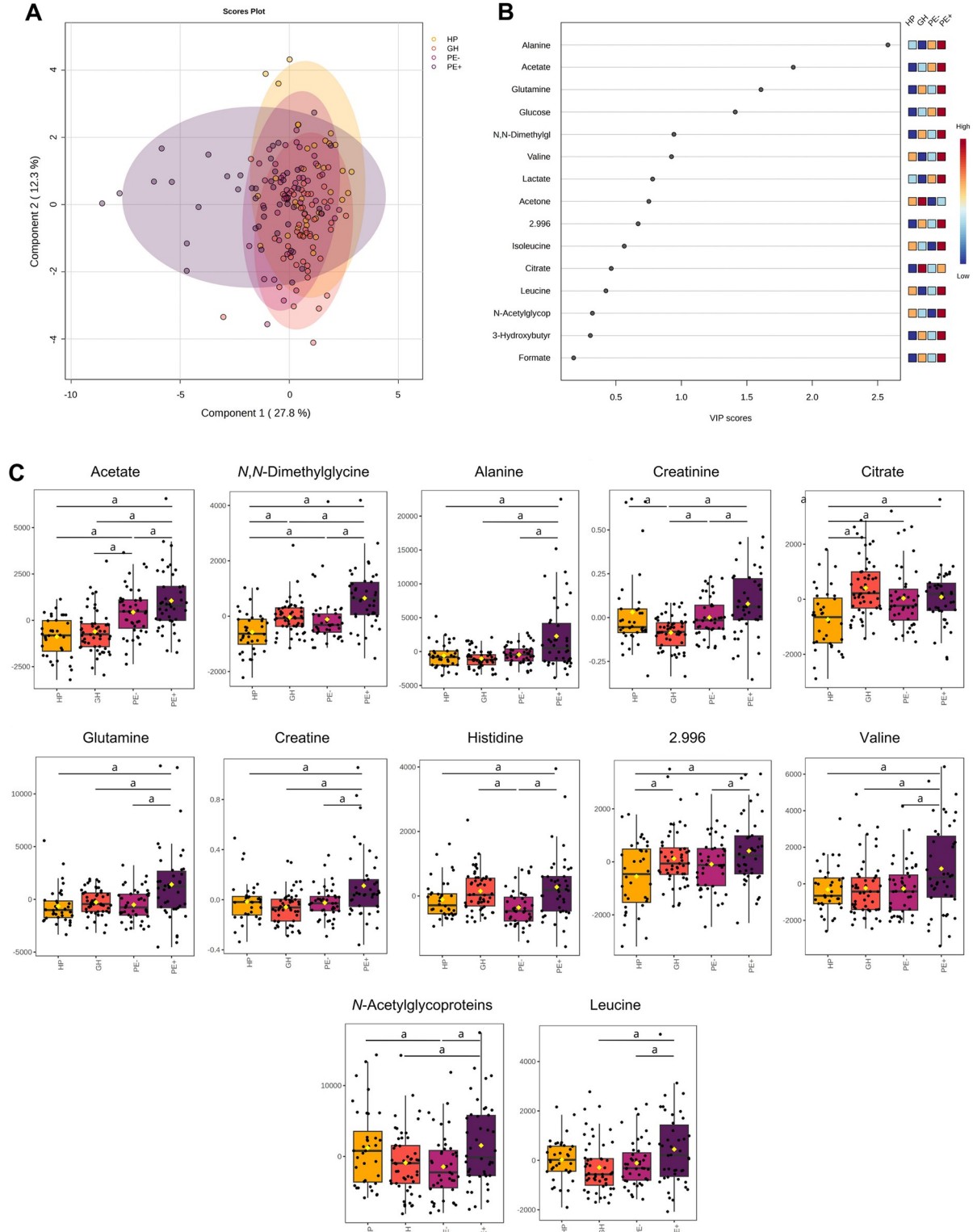

**Fig 2. Metabolomic analysis using the plasma metabolite concentrations identified in healthy pregnant (HP) women, gestational hypertension subjects (GH), preeclampsia patients without severe features (PE–) and with severe features (PE+).** (**A**) 2-dimensional score plot driven by sPLS-DA of the four experimental groups. (**B**) Variable importance in projection (VIP) scores spotlight the 15 most significant metabolites for group differentiation by partial least squares discriminant analysis (PLS-DA). (**C**) Box plots showcasing variations in twelve metabolites with significant differences between the experimental groups, identified in the ¹H-NMR spectra: acetate, *N,N*-

dimethylglycine, alanine, creatinine, citrate, glutamine, creatine, histidine, 2.996, valine, *N*-acetylglycoproteins, and leucine. Data are shown as normalized plasma metabolite concentrations and statistically analyzed by ANOVA and Fisher's LSD. $p < 0.05$ (ª) were considered significant. HP (yellow), GH (orange), PE–(pink), and PE+ (purple).

group. S7 Fig illustrates the loading plots of PLS-DA for all analyzed comparisons, complementing the insights from Figs 2–5 and S4–S6 Figs.

## Metabolites and clinical features correlations

Based on the metabolomics results, it became evident that the metabolites acetate, *N,N*-dimethylglycine, alanine, creatinine, citrate, glutamine, creatine, and valine were of notable significance in differentiating the PE+ cohort from the other groups. To deepen our understanding of the role of metabolites in hypertensive diseases of pregnancy, we conducted correlation analyses focusing on elucidating the relationship between these key metabolites and clinical data. The results of the metabolites that showed significant correlations with the clinical characteristics of the 173 study participants are illustrated in Fig 6. The findings suggest that elevated levels of acetate, *N,N*-dimethylglycine, alanine, citrate, glutamine, and valine are associated with increased blood pressure and worse obstetric and perinatal outcomes. Moreover, a positive correlation was identified between *N,N*-dimethylglycine, alanine, and valine concentrations and poorer end-organ function. Positive correlations were observed between acetate and creatinine levels and renal damage, while glutamine levels were associated with greater hepatic damage. Elevated creatine levels were only linked to worse obstetric and perinatal clinical features. Additionally, creatinine, creatine, and valine were metabolites negatively correlated with gestational age at sampling.

## Discussion

PE+ stands out as a significant complication during pregnancy and the postpartum period. It is marked by a gradual deterioration in the health of both the mother and fetus, ultimately culminating in a poor prognosis [2]. This is the first study to employ $^1$H-NMR for metabolite identification in plasma from PE+ patients. Our primary findings unveiled heightened concentrations of acetate, *N,N*-dimethylglycine, glutamine, alanine, valine, and creatine in the PE + group when compared to both HP and GH groups. Interestingly, only acetate and citrate levels were increased in the PE–group compared to HP. Delving further into the comparative analysis, PE+ exhibited higher concentrations of *N,N*-dimethylglycine, glutamine, alanine, and valine in comparison to the PE–group. The GH group exhibited higher concentrations of citrate and 2.996 compared to the HP group. In addition, elevated levels of specific metabolites, including *N,N*-dimethylglycine, alanine, and valine, were associated with increased blood pressure, worse obstetric outcomes, and poorer end-organ function, particularly renal and hepatic damage. These nuanced findings highlight significant alterations in the metabolic profiles among the HP, GH, PE–, and PE+ groups, providing a unique perspective into the intricate metabolic dynamics associated with different pregnancy-related conditions.

Our research has unveiled elevated levels of glutamine in the plasma of individuals with PE + compared to both HP and PE–groups, consistent with recent evidence of higher glutamine levels in the plasma metabolomic profile of preterm PE [23]. Furthermore, other metabolomics studies have reported increased glutamine levels in placentas of PE+ patients [18], and the serum of early-onset PE patients [15]. Glutamine, synthesized by the enzymatic conversion of glutamate through glutamine synthetase, serves as a crucial precursor for the synthesis of peptides, proteins, amino sugars, nucleic acids, and nucleotides (purines and pyrimidines). It also acts as a carbon source for oxidation in specific cells [24]. In mammalian cells, both glutamine

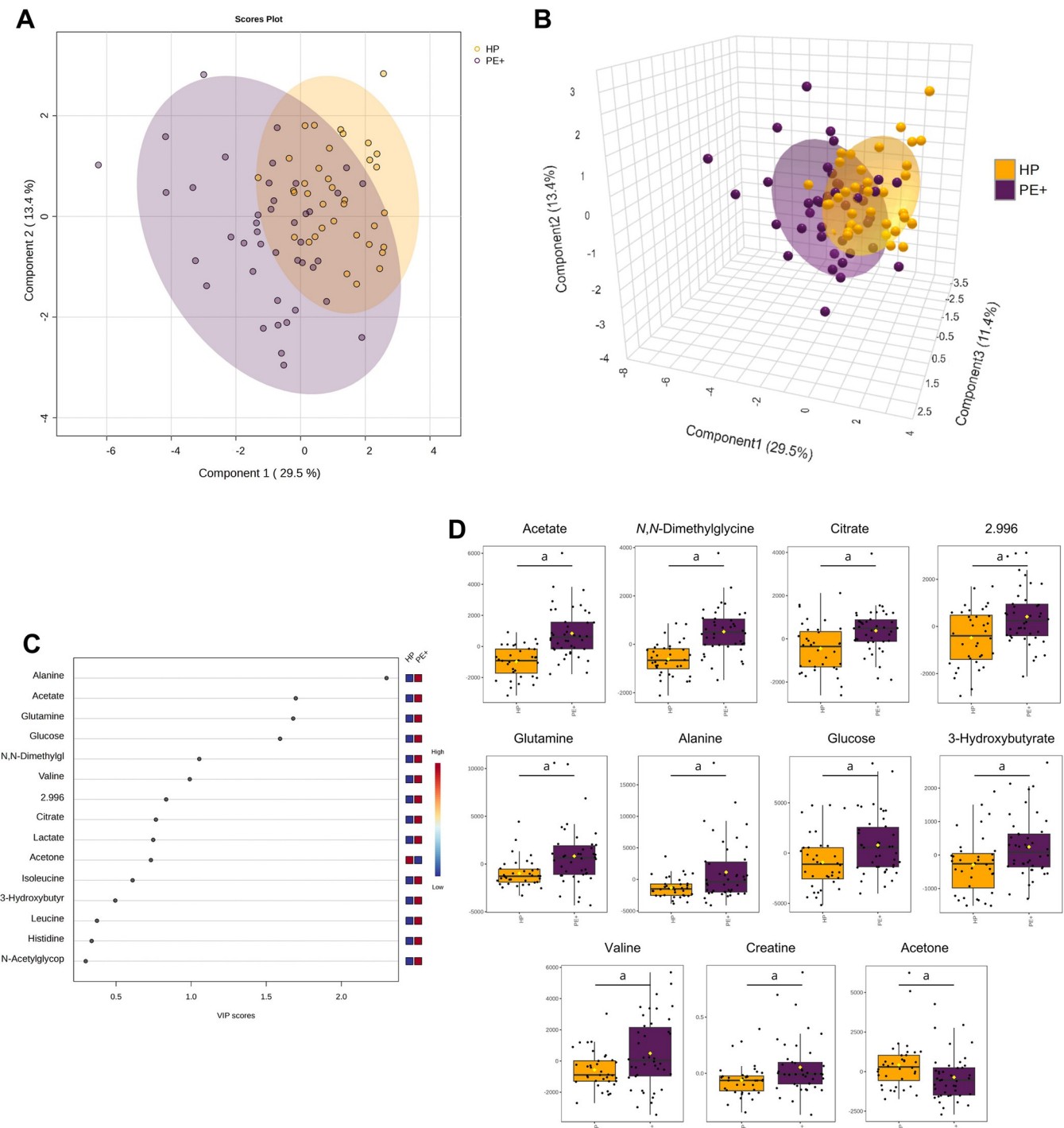

**Fig 3. Metabolomic analysis using the plasma metabolite concentrations identified in healthy pregnant (HP) women and preeclampsia patients with severe features (PE+).** (**A**) 2-dimensional and (**B**) 3-dimensional score plots driven by sPLS-DA of the two experimental groups. (**C**) Variable importance in projection (VIP) scores spotlight the 15 most significant metabolites for group differentiation by PLS-DA. (**D**) Box plots showcasing variations in eleven metabolites with significant differences between HP and PE+ groups, identified in the [1]H-NMR spectra: acetate, *N,N*-dimethylglycine, citrate, 2.996, glutamine, alanine, glucose, 3-hydroxybutyrate, valine, creatine, and acetone. Data are shown as normalized plasma metabolite concentrations and statistically analyzed by Student's unpaired t-test. $p < 0.05$ ([a]) were considered significant.

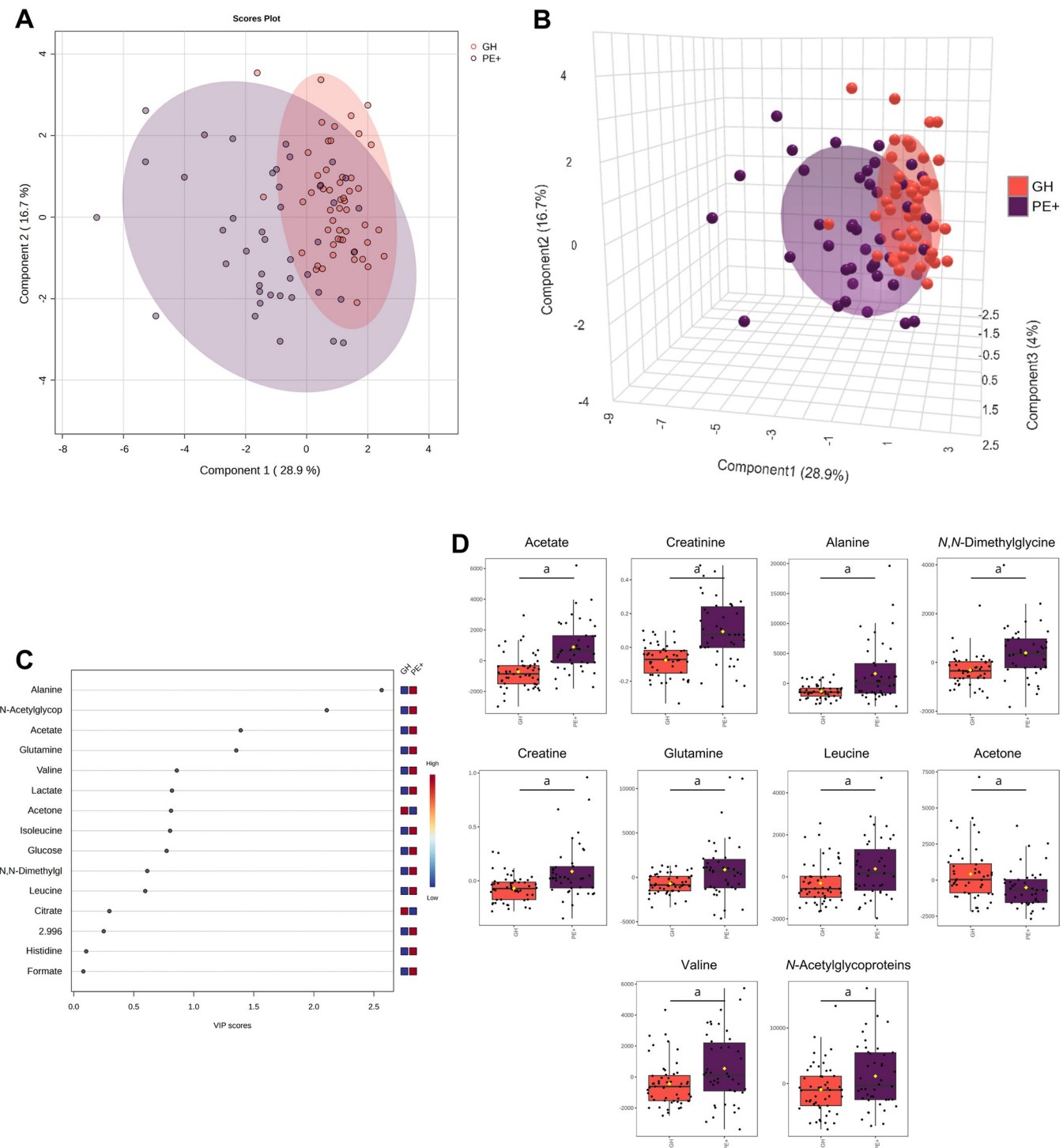

**Fig 4. Metabolomic analysis using plasma metabolite concentrations identified in gestational hypertension subjects (GH) and preeclampsia patients with severe features (PE+).** (**A**) 2-dimensional and (**B**) 3-dimensional score plots driven by sPLS-DA of the two experimental groups. (**C**) Variable importance in projection (VIP) scores spotlight the 15 most significant metabolites for group differentiation by PLS-DA. (**D**) Box plots showcasing variations in eleven metabolites with significant differences between GH and PE+ groups, identified in the $^1$H-NMR spectra: acetate, creatinine, alanine, *N,N*-dimethylglycine, creatine, glutamine, leucine, acetone, valine, and *N*-acetylglycoproteins. Data are shown as normalized plasma metabolite concentrations and statistically analyzed by Student's t-test. $p < 0.05$ ([a]) were considered significant. GH (orange) and PE+ (purple).

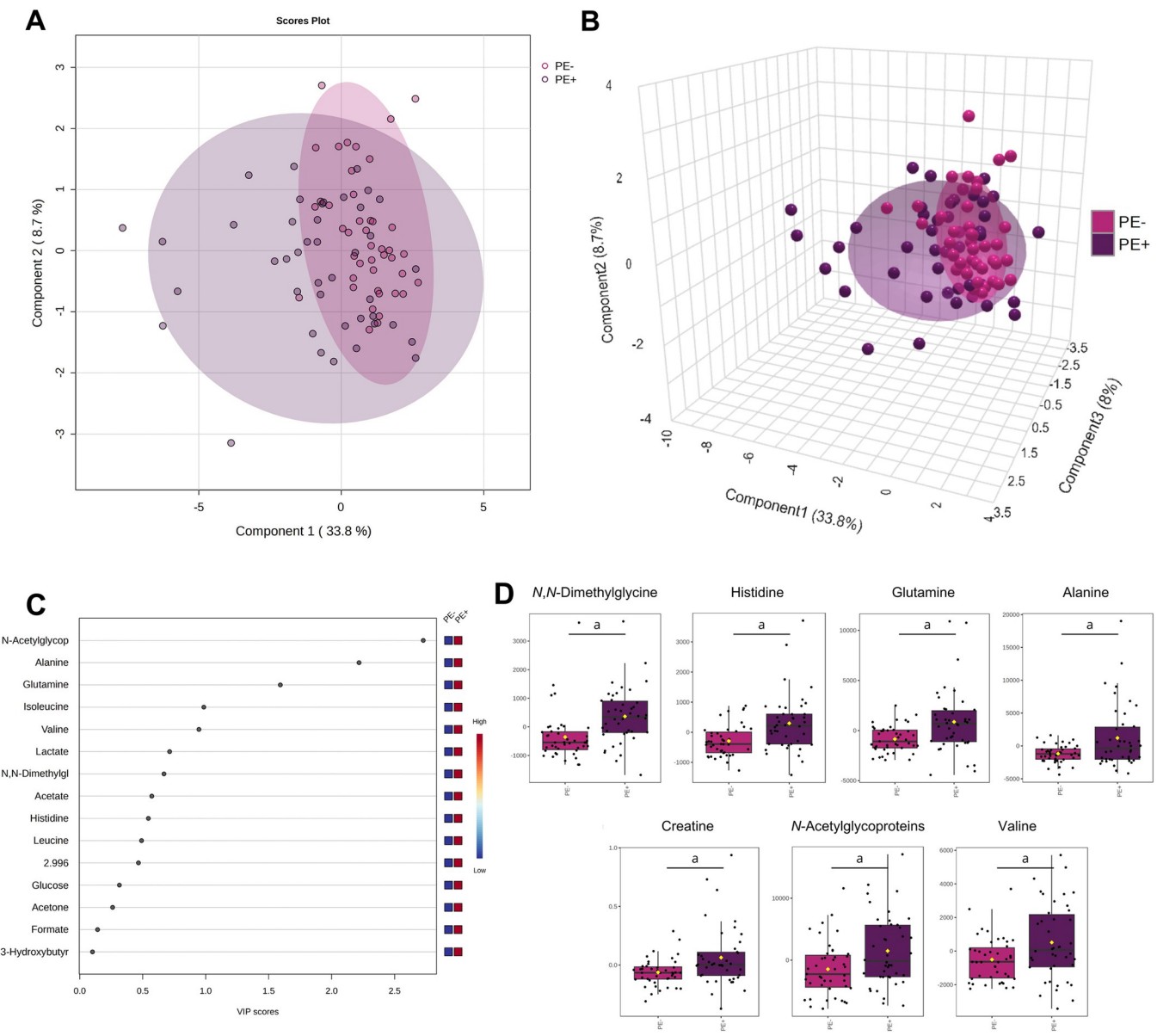

**Fig 5. Metabolomic analysis using plasma metabolite concentrations identified in preeclampsia patients without severe features (PE−) and with severe features (PE+).** (**A**) 2-dimensional and (**B**) 3-dimensional score plots driven by sPLS-DA of the two experimental groups. (**C**) Variable importance in projection (VIP) scores spotlight the 15 most significant metabolites for group differentiation by PLS-DA. (**D**) Box plots showcasing variations in seven metabolites with significant differences between PE−and PE+ groups identified in the $^{1}$H-NMR spectra: *N,N*-dimethylglycine, histidine, glutamine, alanine, creatine, *N*-acetylglycoproteins, and valine. Data are shown as normalized plasma metabolite concentrations and statistically analyzed by Student's t-test. $p < 0.05$ ([a]) were considered significant. PE–(pink) and PE+ (purple).

and histidine are rapidly metabolized into glutamate, a substrate for ornithine production within the urea cycle [24]. Ornithine can be further converted to arginine [24, 25] through NO synthase (NOS) catalysis in endothelial cells, ultimately leading to the production of NO, a potent vasodilator. Although reduced NO bioavailability is a hallmark of endothelial dysfunction in PE [9], the exact impact of elevated glutamine on this pathway in PE+ remains to be clarified. It is noteworthy that, depending on the individual's developmental stage and health, arginine, derived from glutamine/glutamate can be either semi-essential or conditionally

| | GAS (weeks) | SBP (mmHg) | DBP (mmHg) | AST (U/L) | Creat. (mg/dL) | 24 h Pr (mg/24h) | GAD (weeks) | NBW (g) |
|---|---|---|---|---|---|---|---|---|
| Acetate | -0.1 | **0.3** | **0.3** | 0.1 | **0.3** | **0.4** | **-0.4** | **-0.3** |
| *N,N*-dimethylglycine | -0.1 | **0.3** | **0.2** | 0.2 | **0.4** | **0.3** | **-0.3** | **-0.3** |
| Alanine | 0.0 | **0.2** | **0.2** | 0.2 | **0.3** | **0.3** | **-0.2** | **-0.2** |
| Creatinine | **-0.2** | 0.1 | 0.1 | 0.2 | **0.5** | **0.4** | **-0.3** | **-0.3** |
| Citrate | 0.1 | **0.2** | **0.2** | 0.1 | 0.2 | 0.2 | -0.1 | -0.1 |
| Glutamine | -0.1 | **0.2** | **0.2** | **0.2** | 0.1 | 0.3 | **-0.3** | **-0.2** |
| Creatine | **-0.3** | 0.1 | 0.1 | -0.1 | 0.2 | 0.2 | **-0.2** | **-0.2** |
| Valine | **-0.3** | **0.3** | **0.2** | **0.2** | **0.3** | **0.5** | **-0.5** | **-0.5** |

**Fig 6. Correlation between metabolite concentrations and clinical characteristics of the study cohort [a].** 24 h Pr, 24 h proteinuria; AST, aspartate transaminase; Creat., creatinine; DBP, diastolic blood pressure; GAD, gestational age at delivery; NBW, newborn weight; SBP, systolic blood pressure. [a] Values are given as rs Spearman correlation coefficients. Bold values are correlations with significant *p*-values.

essential [26]. Interestingly, glutamine was inversely associated with blood pressure in human subjects without diabetes mellitus or cardiovascular disease [27]. In mice, elevated glutamine levels have been linked to improved glucose tolerance and reduced blood pressure following exogenous supplementation [27]. Additionally, endogenous glutamine has been identified as playing a crucial role in maintaining antioxidant capacity through the reduction of the loss of total and reduced glutathione caused by hydrogen peroxide [28]. However, our findings underscore the higher levels of glutamine in the PE+ group compared to both the HP and PE– groups, as well as a correlation between elevated glutamine levels and increased blood pressure, renal damage, and worse obstetric outcomes across all groups. This may indicate that the elevation of glutamine serves as a compensatory mechanism by the organism to alleviate severe metabolic and oxidative stress-related dysfunctions, as observed in PE+. Nevertheless, the positive correlation between glutamine levels and greater hepatic damage suggests that hepatic dysfunction may be leading to glutamine accumulation. These findings underscore the need for further research to validate this hypothesis and to explore the potential role of glutamine in the pathophysiology of PE+.

In a metabolomics analysis conducted during the first trimester, elevated levels of *N,N*-dimethylglycine were observed in the serum of women with PE [29], aligning with our findings in the PE+ group. Notably, *N,N*-dimethylglycine appeared to be associated with all hypertensive disorders of pregnancy (GH and PE), exhibiting a marked increase in the PE+ group, and confirmed by the positive correlation with increased blood pressure and poorer end-organ function. The observed upregulation potentially indicates that *N,N*-dimethylglycine may play a role in the pathophysiology of these conditions, particularly in more severe manifestations of PE. The involvement of this metabolite is further substantiated by its correlation with unfavorable obstetric and perinatal outcomes. Although information specifically linking *N,N*-dimethylglycine to PE is limited, it is recognized as a byproduct of homocysteine metabolism during the formation of methionine [30]. Recent research explored the prognostic value of increased serum homocysteine concentration in PE, especially during the first trimester of pregnancy, suggesting homocysteine as a predictor of PE [31]. Furthermore, compelling evidence indicates an association between elevated circulating homocysteine levels and adverse pregnancy outcomes, alongside an increased risk of cardiovascular diseases, endothelial dysfunction, and elevated levels of ADMA (asymmetric dimethylarginine, an endogenous inhibitor of NOS) [32–34]. These interconnected findings shed light on the intricate web of associations between *N,N*-dimethylglycine, homocysteine, and the broader spectrum of health outcomes in the context of PE.

Also consistent with earlier metabolomics discoveries in circulating samples of PE [23, 35, 36], our study reinforces the observation that the PE+ group showcases higher levels of creatine when compared to HP, GH, and PE–groups. Originating as a derivative of amino acids, creatine is primarily localized in human skeletal muscle [37]. It undergoes nonenzymatic

degradation producing creatinine, which is subsequently excreted by the kidneys alongside creatine [37]. Importantly, in deleterious conditions such as PE [2], elevated serum creatinine levels are often indicative of potential renal impairment [37]. In our study, we observed a significant correlation between creatinine levels and clinical evidence of kidney dysfunction. Additionally, both elevated creatine and creatinine levels were negatively correlated with gestational age at sampling, suggesting that these metabolites are detected at higher concentrations earlier in pregnancy. Although the gestational age at sampling tends to be earlier and is an inherent characteristic of the PE+ group, this correlation limits our ability to draw definitive conclusions about their role in PE+, emphasizing the need for further investigation into their potential as markers of renal implications associated with this condition.

Circulating levels of alanine in patients with PE have been a subject of conflicting discussion [38–41]. However, our study reveals a distinctive pattern for this metabolite within the groups, illustrating that the PE+ group exhibited higher alanine levels when compared to HP, GH, and PE–groups. Higher alanine concentrations were also related to elevated blood pressure, end-organ damage, and worse obstetric and perinatal clinical features. Alanine, classified as an alpha-amino acid, assumes a crucial role in the glucose-alanine cycle between tissues and the liver, contributing to gluconeogenesis [42]. It actively participates in a reversible transamination reaction with α-oxoglutarate, catalyzed by ALT (alanine aminotransferase). This process generates pyruvate and glutamate, which are then utilized in liver gluconeogenesis [42, 43], thereby maintaining the delicate balance of amino acid metabolism and energy production. Moreover, the levels of alanine during pregnancy may reflect the changes in amino acid concentrations throughout the gestational period. Supporting evidence suggests that alanine levels are augmented in the third trimester [44], which may be associated with dynamic alterations in metabolic demands as pregnancy advances. Notably, ALT is a marker for hepatic injury in PE patients [2]. Interestingly, our metabolomics analysis detected higher glucose levels in the plasma of the PE+ group compared to the HP group. This intriguing interplay suggests that increased levels of alanine within the context of PE+ may also imply disruptions in hepatic function and metabolic homeostasis. These findings prompt further exploration into the intricate connections between alanine metabolism, hepatic injury, and glucose dynamics in those patients.

Acetate (or acetic acid) plays a pivotal role in glycolysis, lipogenesis, and protein acetylation when bound to coenzyme A to form acetyl-CoA [45]. The cell actively maintains low concentrations of free acetic acid due to its acidic properties, preventing potential pH imbalances in cellular contents. A prior study demonstrated a positive correlation between urea levels, systolic and diastolic blood pressure, and elevated serum acetic acid in the PE group [46]. This aligns with our findings, which revealed higher acetate concentrations in both PE+ and PE– groups compared to controls. Furthermore, the elevated acetate levels in PE– in comparison to HP and GH indicate that this metabolite is distinctive to PE. This is further corroborated by the augmented acetate levels in the PE+ group, which aligns with the hypothesis that PE+ represents an exacerbated form of PE. Since short-chain fatty acids (SCFA), including acetic acid, are the primary end products of bacterial metabolism within the intestinal lumen, SCFA levels are highly influenced by the gut microbiota [47]. Alterations in gut microbiota composition are linked to hypertension development during pregnancy [48, 49]. Nevertheless, various factors such as dietary choices, age, antibiotic consumption, and lifestyle influence gut microbiota [49] suggesting that additional pregnancy-related factors may contribute to PE development. Although acetate was not a significant differentiating factor between the PE–and PE+ groups when comparing the two groups alone, its observed upregulation across the PE–group and further in the PE+ group compared to HP and GH groups, in addition to the correlation with

augmented blood pressure, renal damage, and unfavorable obstetric and perinatal outcomes, suggests a potential role for acetate in the progression or severity of PE.

Valine, an essential amino acid requiring dietary intake [50], is implicated in the development of insulin resistance through the regulation by vascular fatty acids transport when present in excess [50]. Insulin resistance is a recognized risk factor for PE [51] potentially leading to diabetes mellitus. However, the elevated valine concentrations in the PE+ group compared to the PE–group do not necessarily indicate insulin resistance. Instead, they may be influenced by the different degrees of obesity among the groups or even the earlier gestational age at sampling. Notably, elevated valine concentrations were also correlated with elevated blood pressure, impaired hepatic and renal functions, and more adverse obstetric and perinatal clinical features, suggesting that valine may play a more complex role in PE development, particularly in the progression to severe features of the disorder. Further studies are essential to clarify these relationships.

The data indicated that *N*-acetylglycoprotein emerged as a distinctive molecular set in comparisons between the PE+ vs GH groups, as well as PE+ vs PE–groups. It was previously identified as an inflammatory biomarker of atherosclerotic cardiovascular disease and a predictor of future risk for hypertension and metabolic syndrome [52]. Moreover, *N*-acetylglycoproteins were elevated in pregnant women with increased BMI [53] and reduced in GH pregnant women compared to healthy controls [54]. Nevertheless, our study did not reveal any notable discrepancies in *N*-acetylglycoprotein levels when comparing our GH and HP groups, despite it being the most important metabolite for group differentiation. Unfortunately, our analysis was unable to identify specific glycoproteins containing *N*-acetyl groups due to signal superimposition from various molecules within the *N*-acetylglycoproteins ppm region, thereby limiting further conclusions.

This study encountered some limitations, including those intrinsic to the study design, related to establishing cause-and-effect relationships between metabolites and hypertensive disorders during pregnancy. Gestational age at sampling varied between the PE+ group and the other groups, which is an inherent characteristic of the PE+ group due to the severity of the disorder, often necessitating preterm delivery. Notably, creatine, creatinine, and valine concentrations were negatively correlated with gestational age at sampling, suggesting that earlier sampling may have contributed to the elevated levels observed in these metabolites, which could be a confounding factor in interpreting their associations with hypertensive disorders. Importantly, no such correlations were found with the other metabolites analyzed. The HP group exhibited lower BMI measurements than the groups of hypertensive disorders. However, this may be influenced by the fact that it was assessed during pregnancy, a period characterized by physiological changes such as edema, which is particularly exacerbated in cases of GH and PE due to increased endothelial dysfunction and capillary permeability [55, 56]. It is therefore plausible that these fluctuations in fluid retention could influence BMI measurements, potentially masking the real extent of obesity in these women. Despite anticipating higher plasma concentrations of creatinine, an established biomarker for renal dysfunction in PE, no significant differences were identified in creatinine levels between HP and both PE groups. In contrast, in the PE+ group and across all groups, significant correlations were observed between creatinine levels assessed by clinical testing and those identified by metabolomics analyses. Additionally, we did not analyze information on dietary intake and some other lifestyle factors that may indeed influence the metabolomic profiles, although all participants were fasting before blood collection to mitigate some of this influence. Caution is advised in extrapolating the findings given the limited patient cohort from a southeastern Brazilian city. Despite this, the strengths of this study lie in well-matched maternal ages and well-characterized experimental groups including HP women with no medical issues, GH subjects with

elevated systolic and diastolic blood pressures, and PE–and PE+ patients with varying degrees of hypertension and proteinuria. The PE+ group presented additional complications, including liver and/or renal impaired functions, along with a lower gestational age at delivery and newborn weight. Rigorous experimental protocols were adhered to during data collection and analysis, providing novel insights into the field of PE+.

The study revealed significant alterations in metabolite profiles among pregnancies affected by hypertensive disorders. These findings may contribute to several clinical applications and personalized treatments, such as risk stratification, optimizing maternal-fetal outcomes, potential for targeted interventions, and personalized therapeutic approaches. Nevertheless, further investigations are essential to strengthen the reliability and applicability of these findings. Future research should explore the metabolite dynamics throughout different stages of pregnancy to elucidate the temporal evolution over the course of gestation.

## Conclusion

In this study, distinct plasma metabolomic profiles were identified among HP subjects and those diagnosed with GH, PE–, and PE+. Notably, findings reveal elevated levels of various plasma metabolites, including amino acids such as glutamine, alanine, and valine, amino acids derivatives such as *N*,*N*-dimethylglycine and creatine, and a fatty acid (acetate), in PE+ patients compared to controls. These observations suggest an exacerbated metabolic disturbance, implicating nitrogen metabolism, methionine, and urea cycles, with indications of renal impairment and hepatic dysfunction in individuals diagnosed with PE+. To corroborate and extend these findings, further investigations are essential. Validation of our results and exploring the intricate associations among the identified metabolites will deepen our understanding of the pathophysiological alterations characteristic of women diagnosed with PE+. A more comprehensive insight into the complexities of PE+ may pave the way for targeted therapeutic interventions.

## Supporting information

**S1 Fig. All NMR spectra were superposed within designated categories (0.0–9.0, and insets 6.5–8.5 ppm).** The spectra were acquired on the Bruker AVANCE III 600 MHz at 25°C and obtained by applying a $^1$H-NMR pulse sequence with a T2 filter (cpmgpr1d). 1, leucine; 2, isoleucine; 3, valine; 4, lactate; 5, alanine; 6, acetate; 7, acetone; 8, 3-hydroxybutyrate; 9, glutamine; 10, citrate; 11, creatine; 12, creatinine; 13, glucose; 14, tyrosine; 15, *N*,*N*-dimethylglycine; 16, histidine; 17, formate; 18, *N*-acetylglycoproteins.
(TIF)

**S2 Fig. Correlation map for $^1$H-$^1$H NMR TOCSY experiment.** 1, leucine; 2, isoleucine; 3, valine; 4, lactate; 5, alanine; 6, acetate; 7, acetone; 8, 3-hydroxybutyrate; 9, glutamine; 10, citrate; 11, creatine; 12, creatinine; 13, glucose; 14, tyrosine; 15, *N*,*N*-dimethylglycine; 16, histidine; 17, formate; 18, *N*-acetylglycoproteins.
(TIF)

**S3 Fig. Scatter plots for correlations between creatinine levels obtained from routine clinical testing and those measured by $^1$H-NMR.** The rs value indicates the Spearman correlation coefficient, whereas the r value indicates the Pearson correlation coefficient.
(TIF)

**S4 Fig. Metabolomic analysis using the plasma metabolite concentrations identified in healthy pregnant (HP) women and gestational hypertension subjects (GH).** (**A**) 2-dimensional and (**B**) 3-dimensional score plots driven by sPLS-DA of the two experimental groups.

(**C**) Variable importance in projection (VIP) scores spotlight the 15 most significant metabolites for group differentiation by PLS-DA. (**D**) Box plots showcasing variations in four metabolites with significant differences between HP and GH groups identified in the [1]H-NMR spectra: citrate, *N,N*-dimethylglycine, creatinine, and 2.996. Data are shown as normalized plasma metabolite concentrations and statistically analyzed by Student's t-test. $p < 0.05$ ([a]) were considered significant. HP (yellow) and GH (orange).
(TIF)

**S5 Fig. Metabolomic analysis using the plasma metabolite concentrations identified in healthy pregnant (HP) women and preeclampsia patients without severe features (PE–).** (**A**) 2-dimensional and (**B**) 3-dimensional score plots driven by sPLS-DA of the two experimental groups. (**C**) Variable importance in projection (VIP) scores spotlights the 15 most significant metabolites for group differentiation by PLS-DA. (**D**) Box plots showcasing variations in two metabolites with significant differences between HP and PE–groups identified in the [1]H-NMR spectra: acetate and citrate. Data are shown as normalized plasma metabolite concentrations and statistically analyzed by Student's t-test. $p < 0.05$ ([a]) were considered significant. HP (yellow) and PE–(pink).
(TIF)

**S6 Fig. Metabolomic analysis using the plasma metabolite concentrations identified in gestational hypertension subjects (GH) and preeclampsia patients without severe features (PE–).** (**A**) 2-dimensional and (**B**) 3-dimensional score plots driven by sPLS-DA of the two experimental groups. (**C**) Variable importance in projection (VIP) scores spotlight the 15 most significant metabolites for group differentiation by PLS-DA. (**D**) Box plots showcasing variations in three significant metabolites between GH and PE–groups as identified in the [1]H-NMR spectra: acetate, histidine, and creatinine. Data are shown as normalized plasma metabolite concentrations and statistically analyzed by Student's t-test. $p < 0.05$ ([a]) were considered significant. GH (orange) and PE–(pink).
(TIF)

**S7 Fig.** Loading plots of the PLS-DA comparing (**A**) healthy pregnant (HP) women, gestational hypertension subjects (GH), and preeclampsia patients without severe features (PE–) and preeclampsia patients with severe features (PE+), (**B**) HP and PE+, (**C**) GH and PE+ (**D**) PE–and PE+, (**E**) HP and GH, (**F**) HP and PE–, (**G**) GH and PE–.
(TIF)

**S1 Table. Clinical, demographic, biochemical, and metabolite data of the participants enrolled in this study.**
(XLSX)

## Acknowledgments

The authors acknowledge and appreciate the valuable contributions of all participants who enrolled in this study and all who aided in the development of this manuscript.

## Author Contributions

**Conceptualization:** Julyane N. S. Kaihara, Valeria Cristina Sandrim.

**Data curation:** Julyane N. S. Kaihara.

**Formal analysis:** Julyane N. S. Kaihara, Fabio Rogerio de Moraes.

**Funding acquisition:** Julyane N. S. Kaihara, Marco G. Alves, Ljubica Tasic, Valeria Cristina Sandrim.

**Investigation:** Julyane N. S. Kaihara.

**Methodology:** Julyane N. S. Kaihara, Fabio Rogerio de Moraes.

**Project administration:** Valeria Cristina Sandrim.

**Resources:** Ljubica Tasic, Valeria Cristina Sandrim.

**Software:** Julyane N. S. Kaihara, Fabio Rogerio de Moraes.

**Supervision:** Ljubica Tasic, Valeria Cristina Sandrim.

**Validation:** Julyane N. S. Kaihara.

**Visualization:** Julyane N. S. Kaihara, Valeria Cristina Sandrim.

**Writing – original draft:** Julyane N. S. Kaihara.

**Writing – review & editing:** Julyane N. S. Kaihara, Fabio Rogerio de Moraes, Priscila Rezeck Nunes, Marco G. Alves, Ricardo C. Cavalli, Ljubica Tasic, Valeria Cristina Sandrim.

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
