## [Decision Letter · Decision Letter 0]

7 May 2024

PONE-D-24-07221Plasma metabolic profile reveals signatures of maternal health during gestational hypertension and preeclampsia without and with severe featuresPLOS ONE

Dear Dr. Sandrim,

Thank you for submitting your manuscript to PLOS ONE. After careful consideration, we feel that it has merit but does not fully meet PLOS ONE’s publication criteria as it currently stands. Therefore, we invite you to submit a revised version of the manuscript that addresses the points raised during the review process.

We look forward to receiving your revised manuscript.

Kind regards,

Anil Bhatia, Ph.D

Academic Editor

PLOS ONE

Journal Requirements:

This work was supported by the Coordenação de Aperfeiçoamento de Pessoal de Nível Superior (CAPES) [grant number 88887.806462/2023-00 (JNSK)] , the Conselho Nacional de Desenvolvimento Científico e Tecnológico (CNPq) [grant numbers 308079/2021-3 (LT) and 308504/2021-6 (VCS)], and the Fundação de Amparo à Pesquisa do Estado de São Paulo (FAPESP) [grant numbers 2018/24069-3 (LT), 2019/07230-8, 2021/12010-7 (VCS), and 2023/08897-1 (JNSK)]. iBiMED is funded by Fundação para a Ciência e a Tecnologia (FCT) [UIDP/04501/2020 and UIDB/04501/2020 (MGA)]. The founders had no role in study design, data collection and analysis, decision to publish, or preparation of the manuscript. 

Reviewers' comments:

Reviewer's Responses to Questions

**Comments to the Author**

1. Is the manuscript technically sound, and do the data support the conclusions?

Reviewer #1: Yes

Reviewer #2: Yes

2. Has the statistical analysis been performed appropriately and rigorously? 

Reviewer #1: Yes

Reviewer #2: Yes

3. Have the authors made all data underlying the findings in their manuscript fully available?

Reviewer #1: Yes

Reviewer #2: Yes

4. Is the manuscript presented in an intelligible fashion and written in standard English?

Reviewer #1: Yes

Reviewer #2: Yes

5. Review Comments to the Author

**Reviewer #1: **Review for Manuscript: “Plasma metabolic profile reveals signatures of maternal health during gestational hypertension and preeclampsia without and with severe features”.

This is an interesting manuscript on an important issue. The study design is retrospective study with 173 subjects. The authors have conducted a study on the metabolomic profile in maternal plasma during pregnancies affected by hypertensive disorders. The authors have used Nuclear magnetic resonance spectroscopy to determine the metabolomic profiles.

I have some recommendations for the authors:

1. The primary weakness of this study is its retrospective design, which may introduce biases and limitations that could affect the reliability of the findings. Authors should acknowledge and discuss these potential limitations in their manuscript.

2. The authors should explain why they have chosen nuclear magnetic resonance spectroscopy and give the advantages.

3. A study flow chart may be helpful for the reader.

4. The authors have concluded that the exacerbated metabolic disturbance discloses renal impairment and hepatic dysfunction, evidenced by elevated levels of creatine and alanine. It would be better to use “may have disclosed”.

5. Body mass index of the healthy subjects is significantly lower than the other three groups. This may have caused a bias while comparing healthy with the other three groups. This should be stated.

6. The authors have collected the blood of preeclamptic women upon diagnosis. The study has a retrospective design. How did the authors get the blood samples?

7. In the discussion section, the authors should address the clinical relevance of their findings. Specifically, they should discuss how their results may be applied in a clinical context.

**Reviewer #2: **The health of expectant mothers and newborns can is seriously compromised by preeclampsia, a pregnancy-specific disease that can worsen over time. Studying the pathways and mechanisms underlying the pathophysiology of severe preeclampsia symptoms and the link with low molecular weight metabolites is an important issue. The topic is relevant because it refers to the health of the pregnant women. The manuscript has an excellent structure and description. The overall paper is organised and well-written. The tables are well-designed and comprehensible. The conclusions are consistent, and they address the main question. The references are appropriate.

I have some remarks:

The abbreviation of gestational hypertension is defined in Line 120 of the article. Before this, the two words are used more than once. I propose to define the abbreviation when you discuss gestational hypertension for the first time, and after that, it is recommended to use the abbreviation. In the tables, the abbreviations are defined many times. The same observation is made about the healthy pregnant abbreviation.

In Line 458, Valine, an essential amino acid requiring dietary intake [20]. This reference does not discuss Valine, so I recommend checking it.

I congratulate the authors for their efforts.

6. PLOS authors have the option to publish the peer review history of their article (what does this mean?). If published, this will include your full peer review and any attached files.

Reviewer #1: No

Reviewer #2: No

---

## [Author Response · Author response to Decision Letter 0]

20 May 2024

Journal: PLOS ONE | ISSN: 1932-6203

Manuscript ID: PONE-D-24-07221

Paper: “Plasma metabolic profile reveals signatures of maternal health during gestational hypertension and preeclampsia without and with severe features”

Response to editor’s comments

- We thank you for providing the additional requirements. The files were revised in accordance with the style requirements for file naming and the manuscript formatting guidelines set out by PLOS ONE.

This work was supported by the Coordenação de Aperfeiçoamento de Pessoal de Nível Superior (CAPES) [grant number 88887.806462/2023-00 (JNSK)] , the Conselho Nacional de Desenvolvimento Científico e Tecnológico (CNPq) [grant numbers 308079/2021-3 (LT) and 308504/2021-6 (VCS)], and the Fundação de Amparo à Pesquisa do Estado de São Paulo (FAPESP) [grant numbers 2018/24069-3 (LT), 2019/07230-8, 2021/12010-7 (VCS), and 2023/08897-1 (JNSK)]. iBiMED is funded by Fundação para a Ciência e a Tecnologia (FCT) [UIDP/04501/2020 and UIDB/04501/2020 (MGA)]. The founders had no role in study design, data collection and analysis, decision to publish, or preparation of the manuscript. 

- The Funding Statement was updated, and the following revised version was included in the cover letter:

“This work was supported by the Coordenação de Aperfeiçoamento de Pessoal de Nível Superior (CAPES) [grant number 88887.806462/2023-00 (JNSK)], the Conselho Nacional de Desenvolvimento Científico e Tecnológico (CNPq) [grant numbers 308079/2021-3 (LT) and 308504/2021-6 (VCS)], and the Fundação de Amparo à Pesquisa do Estado de São Paulo (FAPESP) [grant numbers 2018/24069-3 (LT), 2019/07230-8, 2021/12010-7 (VCS), and 2023/08897-1 (JNSK)]. iBiMED is funded by Fundação para a Ciência e a Tecnologia (FCT) [UIDP/04501/2020 and UIDB/04501/2020 (MGA)]. The founders had no role in study design, data collection and analysis, decision to publish, or preparation of the manuscript. There was no additional external funding received for this study.”

Response to reviewers’ comments

Reviewer #1: Review for Manuscript: “Plasma metabolic profile reveals signatures of maternal health during gestational hypertension and preeclampsia without and with severe features”.

This is an interesting manuscript on an important issue. The study design is retrospective study with 173 subjects. The authors have conducted a study on the metabolomic profile in maternal plasma during pregnancies affected by hypertensive disorders. The authors have used Nuclear magnetic resonance spectroscopy to determine the metabolomic profiles.

I have some recommendations for the authors:

1. The primary weakness of this study is its retrospective design, which may introduce biases and limitations that could affect the reliability of the findings. Authors should acknowledge and discuss these potential limitations in their manuscript.

- We would like to thank the reviewer for their constructive feedback. Upon reevaluation, we have identified that our study can be classified as a cross-sectional case-control design. We apologize for any confusion caused by the earlier misclassification. The necessary adjustments have been made in the Materials and Methods section (line 108). This type of study design has limitations, notably in establishing cause and effect relationships between metabolites and hypertensive disorders during pregnancy (lines 462-464).

2. The authors should explain why they have chosen nuclear magnetic resonance spectroscopy and give the advantages.

- This is a valuable suggestion, and we included a brief explanation in lines 84-89 (Introduction section).

3. A study flow chart may be helpful for the reader.

- We agree that a graphical representation of the study could enhance the reader's understanding and have now incorporated it into the study (Figure 1, lines 108 and 123). 

4. The authors have concluded that the exacerbated metabolic disturbance discloses renal impairment and hepatic dysfunction, evidenced by elevated levels of creatine and alanine. It would be better to use “may have disclosed”.

- We have carefully considered your feedback and have revised the passage accordingly to refine the clarity of our findings (line 40).

5. Body mass index of the healthy subjects is significantly lower than the other three groups. This may have caused a bias while comparing healthy with the other three groups. This should be stated.

- We agree that acknowledging potential biases is important in our analysis. It is worth noting that the BMIs were measured during pregnancy, a period characterized by physiological changes such as edema, particularly in cases of gestational hypertension and preeclampsia. It is possible that these fluctuations could influence BMI measurements. This aspect was highlighted in the Discussion section (lines 465-471) to evidence the potential impact on our comparisons.

6. The authors have collected the blood of preeclamptic women upon diagnosis. The study has a retrospective design. How did the authors get the blood samples?

- We extend our apologies once more for the misclassification of the study design. We reinforce that the samples were collected upon diagnosis and stored in a biobank (– 80 ºC). For the present study, we assessed the samples for conducting the RMN experiments (Material and methods section, line 151).

7. In the discussion section, the authors should address the clinical relevance of their findings. Specifically, they should discuss how their results may be applied in a clinical context.

- It is important to highlight the clinical relevance of our findings. We have incorporated this information into the Discussion section (lines 488-494). We appreciate your suggestions.

Reviewer #2: The health of expectant mothers and newborns can is seriously compromised by preeclampsia, a pregnancy-specific disease that can worsen over time. Studying the pathways and mechanisms underlying the pathophysiology of severe preeclampsia symptoms and the link with low molecular weight metabolites is an important issue. The topic is relevant because it refers to the health of the pregnant women. The manuscript has an excellent structure and description. The overall paper is organised and well-written. The tables are well-designed and comprehensible. The conclusions are consistent, and they address the main question. The references are appropriate.

I have some remarks:

The abbreviation of gestational hypertension is defined in Line 120 of the article. Before this, the two words are used more than once. I propose to define the abbreviation when you discuss gestational hypertension for the first time, and after that, it is recommended to use the abbreviation. In the tables, the abbreviations are defined many times. The same observation is made about the healthy pregnant abbreviation. In Line 458, Valine, an essential amino acid requiring dietary intake [20]. This reference does not discuss Valine, so I recommend checking it. I congratulate the authors for their efforts. 

- We appreciate the reviewer for their valuable contributions to our study. Regarding the first point raised, we have amended and defined all the abbreviations upon their first usage, including within Table 1. Additionally, we believe it is preferable to retain the abbreviation definitions in the titles of images and table in order to ensure their standalone comprehensibility, allowing readers to interpret them independently of the main text. Furthermore, we have updated the valine reference in line 456 (PMID: 26950361), as well as in relevant instances (lines 384, 403, 426, 439).

---

## [Decision Letter · Decision Letter 1]

17 Jul 2024

PONE-D-24-07221R1Plasma metabolic profile reveals signatures of maternal health during gestational hypertension and preeclampsia without and with severe featuresPLOS ONE

Dear Dr.  Sandrim,

Thank you for submitting your manuscript to PLOS ONE. After careful consideration, we feel that it has merit but does not fully meet PLOS ONE’s publication criteria as it currently stands. Therefore, we invite you to submit a revised version of the manuscript that addresses the points raised during the review process.

We look forward to receiving your revised manuscript.

Kind regards,

Anil Bhatia, Ph.D

Academic Editor

PLOS ONE

Journal Requirements:

Reviewers' comments:

Reviewer's Responses to Questions

**Comments to the Author**

1. If the authors have adequately addressed your comments raised in a previous round of review and you feel that this manuscript is now acceptable for publication, you may indicate that here to bypass the “Comments to the Author” section, enter your conflict of interest statement in the “Confidential to Editor” section, and submit your "Accept" recommendation.

Reviewer #3: All comments have been addressed

Reviewer #4: All comments have been addressed

2. Is the manuscript technically sound, and do the data support the conclusions?

Reviewer #3: (No Response)

Reviewer #4: Yes

3. Has the statistical analysis been performed appropriately and rigorously? 

Reviewer #3: Yes

Reviewer #4: Yes

4. Have the authors made all data underlying the findings in their manuscript fully available?

Reviewer #3: Yes

Reviewer #4: Yes

5. Is the manuscript presented in an intelligible fashion and written in standard English?

Reviewer #3: Yes

Reviewer #4: Yes

6. Review Comments to the Author

Reviewer #3: SUMMARY

In their manuscript, Kaihara et al report on an NMR based metabolomics study in plasma samples taken from pregnant women with the following pregnancy outcomes: Healthy Pregnancy (HP), Gestational Hypertension (GH), and preeclampsia with (PE+) and without severe features (PE-). For the non-HP women, samples were taken at time of diagnosis, whereas for the HP women, samples were obtained during routine clinical attendance. Samples were prospectively collected and biobanked prior to their analysis. The study design was a case-control study, with several case groups. The authors report differences between (normalised) the median metabolite levels across the different patient groups studied based on ANOVA analysis. In addition, they use PLA-DA and sPLS-DA to explore the ability of the metabolite profiles to differentiate patient groups and elicit the metabolites which contribute most to the class differentiations. In their reporting and interpretation, the authors first report on a global analysis comparing all patient groups and then focus on comparing / differentiating the PE(+) group from the others. In their discussion, the authors elaborate on extensively on the role of the key differentiating metabolites in their metabolite pathways to corroborate the plausibility of their results, where possible other metabolomics studies are referred to.

Overall, their manuscript is well written and well organised. In the main, the authors have performed and reported on a good piece of research and are commended for this work.

MAJOR ISSUES

Given that metabolite levels were not corrected for possible technical-, like gestation age at sampling, and patient confounders, e.g. BMI, age, smoking (not reported), maternal race/ethnicity (not reported), there is a non-negligent risk of over-interpretation of the results. Particularly, the authors focus a lot of their interpretation on the differences between the PE+ and other patient groups. Whereas it appears that metabolite levels in the PE+ are the most dysregulated as compared to the other patient groups, the gestational age of sampling is also on average 5 weeks earlier than in the other groups, which complicates interpretation (is it (mainly) a gestational age effect or is it a reflection of the presence of severe features?). In this context, the authors should consider refocusing their interpretation on differentiating HP from GH and/or PE- (association with hypertensive disorders of pregnancy/ differential association metabolite profiles between GH and PE). Given the syndromic nature of the hypertensive diseases and the spectrum in their presentation, the PE+ results could then be posited as an exacerbated dysregulation of the metabolite level (and its associated biochemical pathways). In the opinion of the reviewer, interpreting the data in this way sidesteps the gestational age of sampling critique and may add to the robustness of the data interpretation. For instance, the acetate levels (Figure 2) indicate that this metabolite is specific to PE, whereby its upregulated in PE- vs HP & GH; this is confirmed in the PE+, whereby the further upregulation is congruent with the assumption the PE+ is a worse form of PE. Similarly N,N-Dimethylglycine appears to associate with all hypertensive disorders of pregnancy (GH & PE), which its upregulation exacerbated in the PE(+)

MINOR ISSUES

ABSTRACT

1. Line 28: it is suggested to rephrase start of sentenced in line with study design as follows: Collected plasmas from…

2. Lines 29 – 30: it is suggested to rephrase for readability: …preeclampsia without (PE-) and with (PE+) severe features

3. Lines 31 – 43: Depending on whether the authors heed the suggestion as per the Major Issue, this section may be rewritten.

INTRODUCTION

1. Line 72: it is suggested to rephrase for readability: Metabolites, vital components…

2. Line 85: it is suggested to rephrase for readability:… widely used approach for accurately quantifying…

3. Line 86: …Consider adding a reference to contemporary review re. use of 1H-NMR in metabolomics

4. Line 84: it is suggested to rephrase for readability: …. between placentas from pregnancies experiencing PE+ and from normotensive pregnancies.

5. Line 91: it is suggested to rephrase for readability: … to compare the metabolomic profiles ….

6. Lines 92-94: it is suggested to rephrase for readability: Studying the complex of changes… can offer…, assist in establishing … intricate landscape of hypertensive disorders of pregnancy.

7. Line 95: it is suggested to rephrase for readability:…mechanisms of the interrelated conditions GH, PE without and with severe features

8. Line 95-97: it is suggested to improve readability by removing sentence: However… or GH.

9. Lines 98-104: Depending on whether the authors heed the suggestion as per the Major Issue, this section may be rewritten.

MATERIALS AND METHODS

1. Line 146: add time in pregnancy when blood of women with GH was collected

2. Line 147; it is suggested to rephrase for readability:… during routine clinical attendance

3. Line 151: NMR (instead of RMN)

RESULTS

1. Line 209: (Re-)Confirm BMI recorded in patient information is BMI at sampling. Typically, the BMI at begin of pregnancy or pre-pregnancy is considered. BMI at sampling may complicate data

2. Line 216 / TABLE 1: When plotting the creatinine levels as obtained by routine clinical testing vs the NMR creatinine read-outs, there is limited correlation (based on data provided in Supplementary Table 1). Can the authors explain this observation.

3. Lines 244: Inclusion of N-Acetylglycoproteins, clearly an aggregated non-metabolite compound may warrant some further discussion [DISCUSSION].

4. Lines 245 – 259: Depending on whether the authors heed the suggestion as per the Major Issue, this section may be rewritten.

5. Lines 273 – 287: Depending on whether the authors heed the suggestion as per the Major Issue, this section may be rewritten / exchanged by e.g. HP vs GH and/or HP vs PE-, and interpretation of PE+ in these contexts (e.g. by plotting box plots HP vs GH vs PE+ and/or HP vs PE- vs PE+.

6. Line 282 – 287: elaboration on clustering as presented in Fig 3E. By including this clustering analysis, the authors indicate that there is some interesting insight to be gleaned from this clustering. However, the clustering results are not mentioned in the discussion? Based on the clustering presented, some commentary on the apparent subgroups in the PE+ class may be appropriate. Performing clustering within the classes PE+ and HP elicit these subgroups even more.

7. Lines 301 – 310: Depending on whether the authors heed the suggestion as per the Major Issue, this section may be rewritten. Highlighting differences between GH and PE+ has limited relevance when the metabolite levels are not different from HP in the first place.

8. Lines 323 – 331; equivalent comment as in Results - comment # 5

9. Lines 343 – 346: Description of S1 and S2 is better placed at start of results where metabolite identifications by NMR are reported.

10. Lines 346 – 353: Depending on whether the authors heed the suggestion as per the Major Issue, this section may be rewritten.

DISCUSSION

1. Depending on whether the authors heed the suggestion as per the Major Issue, this section may be rewritten; writing itself is good. The authors should consider removing possible over-interpretative sections in the discussion, for instance (non-limiting examples):

a. lines 382 – 386: impact of glutamine on arginine bioavailability etc: health effects loosely attributed to downstream processes

b. Lines 386-389: higher glutamine levels (relative to glutamate; no data available) modulate risks/effects of hypertensive disorders. In their research, Kaihara et al observe an opposite association (again without glutamate data); citing contradictory research without explanation

c. Lines 412 – 421: impairment in renal function is a hallmark of PE, cf. proteinurea is one of the diagnostic criteria; the reviewer would be more interested in the apparent lack of association between NMR creatinine and clinical chemistry creatinine

d. Lines 422-437: Alanine levels may reflect differences in AA trajectories (https://

(doi.org/10.3390/nu13093080) in pregnancy; possibly associated with dynamically changing metabolic needs as per the authors discussion

CONCLUSIONS

1. Depending on whether the authors heed the suggestion as per the Major Issue, this section may be rewritten.

Reviewer #4: The study by Julyane N. S. Kaihara and co-authors provides an excellent analysis of metabolic profiles of preeclampsia women compared with both gestational hypertension and healthy individuals. Moreover, PE women were divided into subgroups according to the severity of symptoms. The paper is accessible, and conclusions are given based on the results. The scientific background of the study is vivid, and the presented data shed light on the pathogenesis of preeclampsia and gestational hypertension.

The given version of the paper was revised after the previous review round. The authors' answers were sufficient, and the text was improved accordingly.

I have only minor suggestions for the authors.

1. The paper would benefit from the correlation analysis of metabolomics results with the clinical characteristics of the study populations.

2. The patient’s characteristics should be moved from the result section to the material and method section.

7. PLOS authors have the option to publish the peer review history of their article (what does this mean?). If published, this will include your full peer review and any attached files.

Reviewer #3: No

Reviewer #4: No

---

## [Author Response · Author response to Decision Letter 1]

22 Aug 2024

Journal: PLOS ONE | ISSN: 1932-6203

Manuscript ID: PONE-D-24-07221R1

Paper: “Plasma metabolic profile reveals signatures of maternal health during gestational hypertension and preeclampsia without and with severe features”

We thank the editor and all the reviewers for your valuable comments on our manuscript entitled “Plasma metabolic profile reveals signatures of maternal health during gestational hypertension and preeclampsia without and with severe features.” In the following section, we provide a point-by-point response to each comment, with the required revisions marked throughout the resubmitted manuscript ‘Revised Manuscript with Track Changes’.

Response to editor’s comments

Journal Requirements:

We have thoroughly reviewed our reference list and can confirm that no retracted papers are included in this manuscript. We have ensured that all cited references are consistent and relevant to our study.

Response to reviewers’ comments

Reviewer #3: SUMMARY

In their manuscript, Kaihara et al report on an NMR based metabolomics study in plasma samples taken from pregnant women with the following pregnancy outcomes: Healthy Pregnancy (HP), Gestational Hypertension (GH), and preeclampsia with (PE+) and without severe features (PE-). For the non-HP women, samples were taken at time of diagnosis, whereas for the HP women, samples were obtained during routine clinical attendance. Samples were prospectively collected and biobanked prior to their analysis. The study design was a case-control study, with several case groups. The authors report differences between (normalised) the median metabolite levels across the different patient groups studied based on ANOVA analysis. In addition, they use PLA-DA and sPLS-DA to explore the ability of the metabolite profiles to differentiate patient groups and elicit the metabolites which contribute most to the class differentiations. In their reporting and interpretation, the authors first report on a global analysis comparing all patient groups and then focus on comparing / differentiating the PE(+) group from the others. In their discussion, the authors elaborate on extensively on the role of the key differentiating metabolites in their metabolite pathways to corroborate the plausibility of their results, where possible other metabolomics studies are referred to.

Overall, their manuscript is well written and well organised. In the main, the authors have performed and reported on a good piece of research and are commended for this work.

MAJOR ISSUES

Given that metabolite levels were not corrected for possible technical-, like gestation age at sampling, and patient confounders, e.g. BMI, age, smoking (not reported), maternal race/ethnicity (not reported), there is a non-negligent risk of over-interpretation of the results. Particularly, the authors focus a lot of their interpretation on the differences between the PE+ and other patient groups. Whereas it appears that metabolite levels in the PE+ are the most dysregulated as compared to the other patient groups, the gestational age of sampling is also on average 5 weeks earlier than in the other groups, which complicates interpretation (is it (mainly) a gestational age effect or is it a reflection of the presence of severe features?). In this context, the authors should consider refocusing their interpretation on differentiating HP from GH and/or PE- (association with hypertensive disorders of pregnancy/ differential association metabolite profiles between GH and PE). Given the syndromic nature of the hypertensive diseases and the spectrum in their presentation, the PE+ results could then be posited as an exacerbated dysregulation of the metabolite level (and its associated biochemical pathways). In the opinion of the reviewer, interpreting the data in this way sidesteps the gestational age of sampling critique and may add to the robustness of the data interpretation. For instance, the acetate levels (Figure 2) indicate that this metabolite is specific to PE, whereby its upregulated in PE- vs HP & GH; this is confirmed in the PE+, whereby the further upregulation is congruent with the assumption the PE+ is a worse form of PE. Similarly N,N-Dimethylglycine appears to associate with all hypertensive disorders of pregnancy (GH & PE), which its upregulation exacerbated in the PE(+).

We appreciate the reviewer’s valuable comments and suggestions regarding the interpretation of our results, particularly concerning the comparisons between PE+ and other patient groups. The emphasis on PE+ in our study was driven by the clinical significance and severity of this condition, with greater risks to both maternal and neonatal health, often necessitating earlier delivery as a clinical intervention. Therefore, the metabolite profile differences observed in PE+ are of particular interest, as they may provide insights into the pathophysiology of this disorder. 

Regarding the gestational age at sampling, we acknowledge the potential confounding effect of gestational age at sampling, but it indeed tends to be earlier and is an inherent characteristic of the PE+ group. We analyzed correlations between metabolite levels and gestational age at sampling, finding that creatine, creatinine, and valine levels were negatively correlated, which could be a confounding factor in interpreting their associations with hypertensive disorders. This limitation has been included in our discussion. However, we believe that the metabolite dysregulation observed in the PE+ group is still of significant value, as no such correlations were found with other metabolites. Furthermore, we have incorporated the reviewer’s suggestions regarding the findings on acetate and N,N-dimethylglycine levels into the Discussion section. We believe that these revisions improved the quality of our manuscript.

MINOR ISSUES

ABSTRACT

1. Line 28: it is suggested to rephrase start of sentenced in line with study design as follows: Collected plasmas from…

2. Lines 29- 30: it is suggested to rephrase for readability: …preeclampsia without (PE-) and with (PE+) severe features

3. Lines 31-43: Depending on whether the authors heed the suggestion as per the Major Issue, this section may be rewritten.

We appreciate the reviewer’s suggestions for improving the abstract. We have incorporated the recommended changes to our manuscript.

INTRODUCTION

1. Line 72: it is suggested to rephrase for readability: Metabolites, vital components…

2. Line 85: it is suggested to rephrase for readability:… widely used approach for accurately quantifying…

3. Line 86: …Consider adding a reference to contemporary review re. use of 1H-NMR in metabolomics

4. Line 84: it is suggested to rephrase for readability: …. between placentas from pregnancies experiencing PE+ and from normotensive pregnancies.

5. Line 91: it is suggested to rephrase for readability: … to compare the metabolomic profiles ….

6. Lines 92-94: it is suggested to rephrase for readability: Studying the complex of changes… can offer…, assist in establishing … intricate landscape of hypertensive disorders of pregnancy.

7. Line 95: it is suggested to rephrase for readability:…mechanisms of the interrelated conditions GH, PE without and with severe features

8. Line 95-97: it is suggested to improve readability by removing sentence: However… or GH.

9. Lines 98-104: Depending on whether the authors heed the suggestion as per the Major Issue, this section may be rewritten.

We have incorporated the recommended changes to our manuscript, including a review regarding the use of ¹H-NMR in metabolomics (PMID: 31252628).

MATERIALS AND METHODS

1. Line 146: add time in pregnancy when blood of women with GH was collected

2. Line 147: it is suggested to rephrase for readability:… during routine clinical attendance

3. Line 151: NMR (instead of RMN)

We have incorporated the suggested changes into the Materials and Methods section of our manuscript.

RESULTS

1. Line 209: (Re-)Confirm BMI recorded in patient information is BMI at sampling. Typically, the BMI at begin of pregnancy or pre-pregnancy is considered. BMI at sampling may complicate data

2. Line 216 / TABLE 1: When plotting the creatinine levels as obtained by routine clinical testing vs the NMR creatinine read-outs, there is limited correlation (based on data provided in Supplementary Table 1). Can the authors explain this observation.

3. Lines 244: Inclusion of N-Acetylglycoproteins, clearly an aggregated non-metabolite compound may warrant some further discussion [DISCUSSION].

4. Lines 245- 259: Depending on whether the authors heed the suggestion as per the Major Issue, this section may be rewritten.

5. Lines 273-287: Depending on whether the authors heed the suggestion as per the Major Issue, this section may be rewritten / exchanged by e.g. HP vs GH and/or HP vs PE-, and interpretation of PE+ in these contexts (e.g. by plotting box plots HP vs GH vs PE+ and/or HP vs PE- vs PE+.

6. Line 282-287: elaboration on clustering as presented in Fig 3E. By including this clustering analysis, the authors indicate that there is some interesting insight to be gleaned from this clustering. However, the clustering results are not mentioned in the discussion? Based on the clustering presented, some commentary on the apparent subgroups in the PE+ class may be appropriate. Performing clustering within the classes PE+ and HP elicit these subgroups even more.

7. Lines 301-310: Depending on whether the authors heed the suggestion as per the Major Issue, this section may be rewritten. Highlighting differences between GH and PE+ has limited relevance when the metabolite levels are not different from HP in the first place.

8. Lines 323-331; equivalent comment as in Results - comment # 5

9. Lines 343-346: Description of S1 and S2 is better placed at start of results where metabolite identifications by NMR are reported.

10. Lines 346-353: Depending on whether the authors heed the suggestion as per the Major Issue, this section may be rewritten.

- In our study, BMI was recorded at the time of blood sampling, as pre-pregnancy measurements were not consistently available. As a result of the patients being referred from local health centers to the University Hospital of FMRP-USP, a tertiary care facility for high-risk pregnancies, data regarding their pre-pregnancy or early pregnancy BMI were unavailable for approximately 40% of the subjects. Additionally, it is possible that the pre-pregnancy BMI may not be a reliable indicator, due to the potential for memory biases to influence the recollection of pre-pregnancy weight information.

- In the PE+ group, concerning the correlation between creatinine levels obtained from routine clinical testing and those measured by NMR, we conducted a Spearman correlation test, given the non-parametric nature of the data. We observed a correlation between the two sets of creatinine levels (routine clinical testing on the y-axis and NMR on the x-axis), and the results are provided in S3A Fig. After excluding the evident outlier, the data distribution became parametric, and the correlation was sustained using the Pearson correlation test (S3B Fig). We included these results in the manuscript (Metabolomics results section) and supplementary material (S3 Fig).

- We have now included a paragraph discussing the N-acetylglycoproteins findings.

- We agree that Fig 3E offers valuable insights into the PE+ group. To gain further understanding, we have conducted additional analyses using a hierarchical clustering algorithm with the metabolite concentration data within the PE+ group, followed by PLS-DA to represent the separation visually. This resulted in the formation of two distinct subgroups, designated as subgroups 1 (with 10 samples) and 2 (with 32 samples), as illustrated in S4 Fig. We generated a cluster dendrogram to identify the samples comprising each subgroup (S5 Fig). Subsequently, to explore potential contributing factors to the observed differences in metabolite concentrations within the PE+ group, we compared clinical, demographic, and biochemical data between the identified subgroups (S2 Table). We observed that while the subgroups are clinically quite similar, there is a trend towards differences in systolic and diastolic blood pressure, and proteinuria between subgroups 1 and 2. However, it is important to note that no international society dedicated to hypertension, obstetrics, or gynecology has established a classification system regarding these characteristics. Moreover, subgroup 1, which shows a trend of higher systolic and diastolic blood pressure, is composed of samples from patients who exhibited elevated levels of specific metabolites, especially leucine, valine, isoleucine, histidine, glutamine, alanine, and glucose as illustrated in the heatmap (Figure 3E).

- We have now moved the description of S1 and S2 Figs to the start of the results section.

DISCUSSION

1. Depending on whether the authors heed the suggestion as per the Major Issue, this section may be rewritten; writing itself is good. The authors should consider removing possible over-interpretative sections in the discussion, for instance (non-limiting examples):

a. lines 382-386: impact of glutamine on arginine bioavailability etc: health effects loosely attributed to downstream processes

b. Lines 386-389: higher glutamine levels (relative to glutamate; no data available) modulate risks/effects of hypertensive disorders. In their research, Kaihara et al observe an opposite association (again without glutamate data); citing contradictory research without explanation

c. Lines 412-421: impairment in renal function is a hallmark of PE, cf. proteinurea is one of the diagnostic criteria; the reviewer would be more interested in the apparent lack of association between NMR creatinine and clinical chemistry creatinine

d. Lines 422-437: Alanine levels may reflect differences in AA trajectories (https://doi.org/10.3390/nu13093080) in pregnancy; possibly associated with dynamically changing metabolic needs as per the authors discussion

We have reviewed the mentioned sections regarding glutamine and revised the discussion to focus more directly on the findings of our study, emphasizing that while the connections between glutamine levels and the disorder are plausible, they require further research for confirmation. To avoid potential confusion, we also removed the information on the relative levels of glutamine and glutamate, and instead only refer to the results of glutamine levels. Furthermore, we have removed the reference to the genetic deletion of glutamine synthetase. As addressed earlier, we conducted Spearman and Pearson (when excluding the outlier) correlations tests and there is indeed an association between NMR creatinine and clinical testing creatinine levels, which we have included in the manuscript. Additionally, we have included a reference regarding alanine levels during pregnancy.

CONCLUSIONS

1. Depending on whether the authors heed the suggestion as per the Major Issue, this section may be rewritten.

 We appreciate the reviewer's suggestions and comprehend the rationale behind them. However, the manuscript has already been subjected to a thorough review process and has been accepted by two previous reviewers with a specific focus on PE+. A change in the main focus of the article at this point could result in a notable shift in the direction and emphasis of the research, which could conflict with previously accepted findings from peer reviewers. Then, as the major revisions were not incorporated, the conclusion section remains unchanged.

Reviewer #4: The study by Julyane N. S. Kaihara and co-authors provides an excellent analysis of metabolic profil

---

## [Decision Letter · Decision Letter 2]

25 Sep 2024

PONE-D-24-07221R2Plasma metabolic profile reveals signatures of maternal health during gestational hypertension and preeclampsia without and with severe featuresPLOS ONE

Dear Dr. Sandrim,

Thank you for submitting your manuscript to PLOS ONE. After careful consideration, we feel that it has merit but does not fully meet PLOS ONE’s publication criteria as it currently stands. Therefore, we invite you to submit a revised version of the manuscript that addresses the points raised during the review process.

We look forward to receiving your revised manuscript.

Kind regards,

Anil Bhatia, Ph.D

Academic Editor

PLOS ONE

Journal Requirements:

Reviewers' comments:

Reviewer's Responses to Questions

**Comments to the Author**

1. If the authors have adequately addressed your comments raised in a previous round of review and you feel that this manuscript is now acceptable for publication, you may indicate that here to bypass the “Comments to the Author” section, enter your conflict of interest statement in the “Confidential to Editor” section, and submit your "Accept" recommendation.

Reviewer #3: (No Response)

Reviewer #4: All comments have been addressed

2. Is the manuscript technically sound, and do the data support the conclusions?

Reviewer #3: Yes

Reviewer #4: Yes

3. Has the statistical analysis been performed appropriately and rigorously? 

Reviewer #3: Yes

Reviewer #4: Yes

4. Have the authors made all data underlying the findings in their manuscript fully available?

Reviewer #3: Yes

Reviewer #4: Yes

5. Is the manuscript presented in an intelligible fashion and written in standard English?

Reviewer #3: Yes

Reviewer #4: Yes

6. Review Comments to the Author

Reviewer #3: SUMMARY

In their manuscript, Kaihara et al report on an NMR based metabolomics study in plasma samples taken from pregnant women with the following pregnancy outcomes: Healthy Pregnancy (HP), Gestational Hypertension (GH), and preeclampsia with (PE+) and without severe features (PE-). For the non-HP women, samples were taken at time of diagnosis, whereas for the HP women, samples were obtained during routine clinical attendance. Samples were prospectively collected and biobanked prior to their analysis. The study design was a case-control study, with several case groups. The authors report differences between the (normalised) median metabolite levels across the different patient groups studied based on ANOVA analysis and pairwise group analyses. In addition, they use PLA-DA and sPLS-DA to explore the ability of the metabolite profiles to differentiate patient groups and elicit the metabolites contributing most to the class differentiations. In their reporting and interpretation, the authors first report on a global analysis comparing all patient groups and then focus on comparing / differentiating the PE(+) group from the others. In their discussion, the authors elaborate on extensively on the role of the key differentiating metabolites in their metabolite pathways to corroborate the plausibility of their results, where possible other metabolomics studies are referred to.

Overall, upon revision, their manuscript remains well written and well organised. In the main, the authors have performed and reported on a good piece of research.

MAJOR ISSUES

None

MINOR ISSUES

Notwithstanding, the good writing and logical organisation of the manuscript, the reviewer regrets the authors didn’t take the opportunity to reorganize their manuscript and to either posit PE(+) to be on a HP > GH > PE(-) > PE(+) spectrum. By adopting a PE(+) centric view on the study, the authors missed the opportunity discussing whether metabolites can differentiate HP from GH and/or PE and/or PE(+). Having all these subgroups represented in one of the strong appeals of this research. As a result of the authors choice, the manuscript lacks a clear narrative and clear conclusions. It remains unclear what metabolites are adding true value in the understanding of the hypertensive diseases of pregnancy. This is obvious in the metabolites the authors chose to elaborate upon in the discussion.

The reviewer had hoped that the authors would considerably prune their manuscript based on the feedback as provided on the previous version, unfortunately this advice was not heeded. For instance, the authors added a lot of results with regards to the presence of sub-clusters in the PE(+) group. Whereas this work was triggered by the reviewer (and is potentially informative), the clustering and extra results are not featuring in the discussion. If results are not informative in the context of the study (cf. they don’t feature in the discussion), it is better to remove the clustering analysis altogether rather than adding more to it. Same remains true for the contextualisation of the metabolites in the discussion, the contextualisation remains unwieldy leaving the reader with little additional insight as a result of the study.

In summary, the researchers did a good piece of research, but -in the reviewer’s opinion- remained to keen to put too much in the paper making the dissemination of their research less effective than it could be.

ABSTRACT

1. OK

INTRODUCTION

1. Line 52: replace “necessitates” by “often necessitates”

2. Line 61: replace "premature termination of the pregnancy” by “iatrogenic premature delivery” to avoid confusion with abortion

3. Line 70: replace “as accompanied by” with “together with”

Lines 71-72: replace "the intricate landscape of” by “the complexity of”

Line 96-101: too verbose, simplify by 1) removal of “Studying…pregnancy” 2) Add “Studying the levels of metabolites across the different hypertensive disorders of pregnancy may provide deeper insights into…” 3) remove “it underscores the….PE+. Thus” 4) Restart: In this study our primary…

4. Line 107: Rephrase” Subsequently, we investigated the presence of correlations…

MATERIALS AND METHODS

1. Line 191: Add : “ For pairwise group comparisons, post-hoc tests…. “

2. Lines 211-217: Suggestion to temove clustering section

RESULTS

1. Suggestion to remove lines 310 – 315: “Moreover,… clusters”

2. Suggestion in Fig 3: remove panel E

3. Suggestion to remove Description of Figure 3: panel E, lines 325 -327, remove all affiliated supplementary results.

4. Suggestion to Remove lines 328-346

DISCUSSION

1. OK

CONCLUSIONS

1. OK

Reviewer #4: All the questions given to the authors has been adressed accordingly. I have no more questions.

7. PLOS authors have the option to publish the peer review history of their article (what does this mean?). If published, this will include your full peer review and any attached files.

Reviewer #3: No

Reviewer #4: **Yes: **Maciej Zieliński

---

## [Author Response · Author response to Decision Letter 2]

3 Oct 2024

Journal: PLOS ONE | ISSN: 1932-6203

Manuscript ID: PONE-D-24-07221R2

Paper: “Plasma metabolic profile reveals signatures of maternal health during gestational hypertension and preeclampsia without and with severe features”

- We thank the editor and all the reviewers for your valuable comments on our manuscript entitled “Plasma metabolic profile reveals signatures of maternal health during gestational hypertension and preeclampsia without and with severe features.” In the following section, we respond to each comment, with the required revisions marked throughout the resubmitted manuscript ‘Revised Manuscript with Track Changes’.

Response to editor’s comments

Journal Requirements:

- We have thoroughly reviewed our reference list and can confirm that no retracted papers are included in this manuscript. We have ensured that all cited references are consistent and relevant to our study.

Response to reviewers’ comments

Reviewer #3: SUMMARY

In their manuscript, Kaihara et al report on an NMR based metabolomics study in plasma samples taken from pregnant women with the following pregnancy outcomes: Healthy Pregnancy (HP), Gestational Hypertension (GH), and preeclampsia with (PE+) and without severe features (PE-). For the non-HP women, samples were taken at time of diagnosis, whereas for the HP women, samples were obtained during routine clinical attendance. Samples were prospectively collected and biobanked prior to their analysis. The study design was a case-control study, with several case groups. The authors report differences between the (normalised) median metabolite levels across the different patient groups studied based on ANOVA analysis and pairwise group analyses. In addition, they use PLA-DA and sPLS-DA to explore the ability of the metabolite profiles to differentiate patient groups and elicit the metabolites contributing most to the class differentiations. In their reporting and interpretation, the authors first report on a global analysis comparing all patient groups and then focus on comparing / differentiating the PE(+) group from the others. In their discussion, the authors elaborate on extensively on the role of the key differentiating metabolites in their metabolite pathways to corroborate the plausibility of their results, where possible other metabolomics studies are referred to.

Overall, upon revision, their manuscript remains well written and well organised. In the main, the authors have performed and reported on a good piece of research.

MAJOR ISSUES

None

MINOR ISSUES

Notwithstanding, the good writing and logical organisation of the manuscript, the reviewer regrets the authors didn’t take the opportunity to reorganize their manuscript and to either posit PE(+) to be on a HP > GH > PE(-) > PE(+) spectrum. By adopting a PE(+) centric view on the study, the authors missed the opportunity discussing whether metabolites can differentiate HP from GH and/or PE and/or PE(+). Having all these subgroups represented in one of the strong appeals of this research. As a result of the authors choice, the manuscript lacks a clear narrative and clear conclusions. It remains unclear what metabolites are adding true value in the understanding of the hypertensive diseases of pregnancy. This is obvious in the metabolites the authors chose to elaborate upon in the discussion.

The reviewer had hoped that the authors would considerably prune their manuscript based on the feedback as provided on the previous version, unfortunately this advice was not heeded. For instance, the authors added a lot of results with regards to the presence of sub-clusters in the PE(+) group. Whereas this work was triggered by the reviewer (and is potentially informative), the clustering and extra results are not featuring in the discussion. If results are not informative in the context of the study (cf. they don’t feature in the discussion), it is better to remove the clustering analysis altogether rather than adding more to it. Same remains true for the contextualisation of the metabolites in the discussion, the contextualisation remains unwieldy leaving the reader with little additional insight as a result of the study.

In summary, the researchers did a good piece of research, but -in the reviewer’s opinion- remained to keen to put too much in the paper making the dissemination of their research less effective than it could be.

- Thank you for your constructive and helpful feedback. Our primary focus on the PE+ group, as previously stated, was driven by its significant impact on maternal health and the severity of its clinical consequences, as well as the importance in maintaining a cohesive and concise manuscript. We agree that the incorporation of the heatmap and the PE+ subgroup analysis slightly deviated from this focus, and then we have removed these analyses from the manuscript. Furthermore, we chose to keep the spotlight on PE+ while making minor amendments throughout the text to improve its overall clarity and quality. We hope the revised version is more suitable.

ABSTRACT

1. OK

INTRODUCTION

1. Line 52: replace “necessitates” by “often necessitates”

2. Line 61: replace "premature termination of the pregnancy” by “iatrogenic premature delivery” to avoid confusion with abortion

3. Line 70: replace “as accompanied by” with “together with”

Lines 71-72: replace "the intricate landscape of” by “the complexity of”

Line 96-101: too verbose, simplify by 1) removal of “Studying…pregnancy” 2) Add “Studying the levels of metabolites across the different hypertensive disorders of pregnancy may provide deeper insights into…” 3) remove “it underscores the….PE+. Thus” 4) Restart: In this study our primary…

4. Line 107: Rephrase” Subsequently, we investigated the presence of correlations…

MATERIALS AND METHODS

1. Line 191: Add : “ For pairwise group comparisons, post-hoc tests…. “

2. Lines 211-217: Suggestion to temove clustering section

RESULTS

1. Suggestion to remove lines 310 – 315: “Moreover,… clusters”

2. Suggestion in Fig 3: remove panel E

3. Suggestion to remove Description of Figure 3: panel E, lines 325 -327, remove all affiliated supplementary results.

4. Suggestion to Remove lines 328-346

DISCUSSION

1. OK

CONCLUSIONS

1. OK

- We have incorporated these suggestions into our manuscript. To maintain the focus on PE+, the heatmap and clustering analyses related to the subgroups have been removed.

Reviewer #4: All the questions given to the authors has been adressed accordingly. I have no more questions.

- We appreciate the reviewer's valuable suggestions and contribution to improving our paper.

---

## [Decision Letter · Decision Letter 3]

5 Nov 2024

Plasma metabolic profile reveals signatures of maternal health during gestational hypertension and preeclampsia without and with severe features

PONE-D-24-07221R3

Dear Dr. Sandrim,

We’re pleased to inform you that your manuscript has been judged scientifically suitable for publication and will be formally accepted for publication once it meets all outstanding technical requirements.

Kind regards,

Anil Bhatia, Ph.D

Academic Editor

PLOS ONE

Additional Editor Comments (optional):

Reviewers' comments:

Reviewer's Responses to Questions

**Comments to the Author**

1. If the authors have adequately addressed your comments raised in a previous round of review and you feel that this manuscript is now acceptable for publication, you may indicate that here to bypass the “Comments to the Author” section, enter your conflict of interest statement in the “Confidential to Editor” section, and submit your "Accept" recommendation.

Reviewer #3: All comments have been addressed

Reviewer #4: All comments have been addressed

2. Is the manuscript technically sound, and do the data support the conclusions?

Reviewer #3: (No Response)

Reviewer #4: Yes

3. Has the statistical analysis been performed appropriately and rigorously? 

Reviewer #3: Yes

Reviewer #4: Yes

4. Have the authors made all data underlying the findings in their manuscript fully available?

Reviewer #3: Yes

Reviewer #4: Yes

5. Is the manuscript presented in an intelligible fashion and written in standard English?

Reviewer #3: Yes

Reviewer #4: Yes

6. Review Comments to the Author

Reviewer #3: SUMMARY

In their manuscript, Kaihara et al report on an NMR based metabolomics study in plasma samples taken from pregnant women with the following pregnancy outcomes: Healthy Pregnancy (HP), Gestational Hypertension (GH), and preeclampsia with (PE+) and without severe features (PE-). For the non-HP women, samples were taken at time of diagnosis, whereas for the HP women, samples were obtained during routine clinical attendance. Samples were prospectively collected and biobanked prior to their analysis. The study design was a case-control study, with several case groups. The authors report differences between the (normalised) median metabolite levels across the different patient groups studied based on ANOVA analysis and pairwise group analyses. In addition, they use PLA-DA and sPLS-DA to explore the ability of the metabolite profiles to differentiate patient groups and elicit the metabolites contributing most to the class differentiations. In their reporting and interpretation, the authors first report on a global analysis comparing all patient groups and then focus on comparing / differentiating the PE(+) group from the others. In their discussion, the authors elaborate extensively on the role of the key differentiating metabolites in their metabolite pathways to corroborate the plausibility of their results.

MAJOR ISSUES

none

MINOR ISSUES

ABSTRACT

1. OK

INTRODUCTION

1. OK

MATERIALS AND METHODS

1. OK

RESULTS

1. Line 274-275: Replace “all of which were significantly different between all groups.” by “all of which had significantly different levels in at least one patient group”

2. Line 483-484: Remove “Despite this, certain gut bacteria can produce and excrete acetic acid.” � this info is repeated later on in the same paragraph

DISCUSSION

1. OK

CONCLUSIONS

1. Line: 606-607: Replace” This comprehensive insight may pave the way for targeted therapeutic interventions, offering promising avenues for addressing the complexities of this condition.” By “A more comprehensive insight into the complexities of PE+ may pave the way for targeted therapeutic interventions.”

Reviewer #4: The authors have fully addressed the reviewers' questions.

7. PLOS authors have the option to publish the peer review history of their article (what does this mean?). If published, this will include your full peer review and any attached files.

Reviewer #3: No

Reviewer #4: No

---

## [Editor Report · Acceptance letter]

15 Nov 2024

PONE-D-24-07221R3 

PLOS ONE

Dear Dr. Sandrim, 

I'm pleased to inform you that your manuscript has been deemed suitable for publication in PLOS ONE. Congratulations! Your manuscript is now being handed over to our production team.

Kind regards, 

on behalf of

Dr. Anil Bhatia 

Academic Editor

PLOS ONE